# Semantic Image Synthesis with Unconditional Generator

JungWoo Chae[12*]  Hyunin Cho[1*]  Sooyeon Go[1]   Kyungmook Choi[1]    Youngjung Uh[1†]

[1]Yonsei Unviersity, Seoul, South Korea
[2]LG CNS AI Research, Seoul, South Korea
{cjwyonsei, hyunin9528, sooyeon8658, kyungmook.choi, yj.uh}@yonsei.ac.kr
{cjwoolgcns}@lgcns.com

## Abstract

Semantic image synthesis (SIS) aims to generate realistic images that match given semantic masks. Despite recent advances allowing high-quality results and precise spatial control, they require a massive semantic segmentation dataset for training the models. Instead, we propose to employ a pre-trained unconditional generator and rearrange its feature maps according to proxy masks. The proxy masks are prepared from the feature maps of random samples in the generator by simple clustering. The feature rearranger learns to rearrange original feature maps to match the shape of the proxy masks that are either from the original sample itself or from random samples. Then we introduce a semantic mapper that produces the proxy masks from various input conditions including semantic masks. Our method is versatile across various applications such as free-form spatial editing of real images, sketch-to-photo, and even scribble-to-photo. Experiments validate advantages of our method on a range of datasets: human faces, animal faces, and buildings. Code will be available online soon.

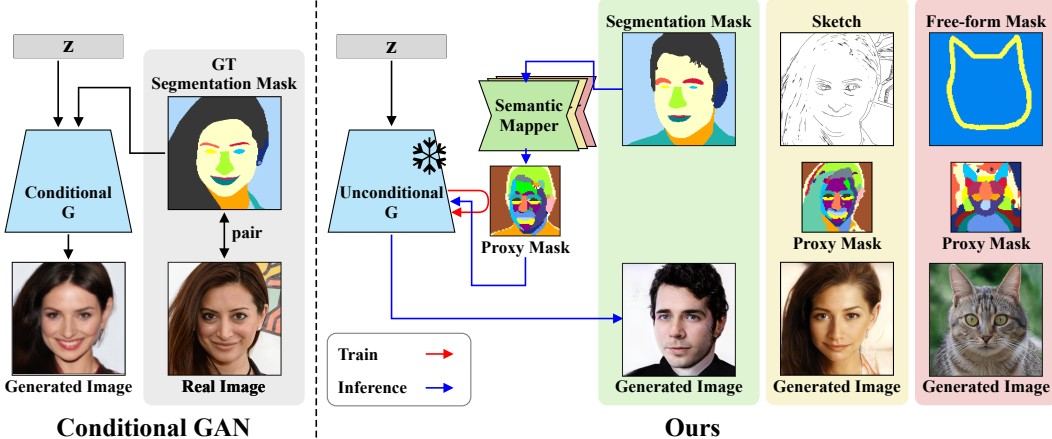

Figure 1: **Conditional GANs vs Ours** (Left) Conditional GANs train a generator using expensive pairs of semantic masks and images. (Right) On the other hand, our method does not require a large number of image-mask pairs in training process and uses a pre-trained unconditional generator for semantic image synthesis. Furthermore, it accommodates various types of inputs such as sketches or even simpler scribbles.

---

*Equal contribution.

†Corresponding author

37th Conference on Neural Information Processing Systems (NeurIPS 2023).

# 1 Introduction

Semantic image synthesis (SIS) aims to synthesize images according to given semantic masks. Its ability to reflect the spatial arrangement of the semantic masks makes it a powerful tool for various applications such as scene generation[21, 25] and photorealistic image editing[34]. However, it requires abundant pairs of images and per-pixel class annotations which are laborious and error-prone. If we want to change the granularity of control, we need to re-annotate the images and re-train the model. E.g., a network pretrained on human faces and their semantic segmentation masks cannot be used for generating human faces from simple scribbles and requires re-training on scribble annotation.

A few studies [20, 4] have attempted to bypass this issue by employing pretrained unconditional generators for image manipulation without the need for annotations. These approaches, however, struggle to allow detailed manipulation, presenting challenges for users desiring to customize generated images. Furthermore, users have to manually identify relevant clusters and tweak feature maps, leading to burdensome manipulation process.

In response to these challenges, we introduce a novel method for semantic image synthesis: rearranging feature maps of a pretrained *unconditional* generator according to given semantic masks. We break down our problem into two sub-problems to accomplish our objectives: rearranging and preparing supervision for the rearrangement. First, we design a rearranger that produces feature maps rearranged according to inputs by aggregating features through attention mechanism. Second, we prepare randomly generated samples, their feature maps, and spatial clustering. Note that the samples are produced online during training thus it does not require additional storage. This approach assumes that we have two random samples and their intermediate feature maps from the generator. The rearranger learns to rearrange the feature maps of one sample to match the semantic spatial arrangement of the other sample.

However, a semantic gap may arise when the generator receives an input mask. As shown in Figure 1(right), this gap is manifested in the discrepancy between the input segmentation mask and the proxy mask. The proxy masks are the arrangement of feature maps which are derived from the feature maps in the generator through simple clustering. We solve this perception difference by training a semantic mapper that can convert an input mask to the proxy mask.

Our proposed method offers several key advantages. Not only does it significantly reduce the training time and the burden of manual annotation, but it also leverages pretrained GANs to yield high-quality synthesized images. Additionally, once we train the rearranger, the need to only train the semantic mapper, even when the segmentation configuration changes, makes our approach computationally efficient compared to methods requiring retraining of the entire model. Moreover, it is compatible with other latent editing techniques such as StyleCLIP[22].

As our method operates through feature maps rearrangement, it enables free-form mask spatial editing with fewer constraints on the mask's shape. Furthermore, our semantic mapper accepts different types of input conditions such as HED [37] and Canny Edges. Therefore, our approach can be applied to various applications like sketch-to-photo and scribble-to-photo transformations.

We demonstrate pixel-level content creation with a pretrained generator on several datasets including CelebAMask-HQ, LSUN Church, and LSUN Bedroom. We also present real image editing examples where the source image is edited to conform to the shape of the target image or target mask. Lastly, our method achieves noticeably higher mIOU scores and significantly lower FID than prior works. Furthermore, our method outperforms the baselines in both qualitative and quantitative measures.

# 2 Related Work

**Semantic image synthesis**    Semantic Image Synthesis (SIS) is a specialized type of conditional image generation that can accurately mirror user intentions at pixel level [7, 33, 40, 17, 27, 35, 21, 47, 25, 26]. Over time, the field has advanced significantly, improving controllability and thus enabling the production of diverse, high-quality results. These advancements also allow for the adjustment of the style of individual class components based on semantic masks. Like preceding GAN methods, the application of diffusion in SIS also facilitates the alteration of image style under specific conditions [34]. More recently, diffusion-based conditional image generation has taken this further, creating high-quality images using semantic masks and other inputs like edge maps and user

scribbles [6, 31, 32, 24, 42]. Despite their benefits, SIS methods are resource-intensive, requiring many image-label pairs during learning. Moreover, any changes in annotations necessitate a complete retraining of the model, leading to inefficiencies. Unlike these methods, our approach uses pairs of generator feature maps and the corresponding generated images as data, effectively eliminating the need for expensive image-label pair datasets. As a result, our method provides significant cost and time savings compared to previous methods.

**Feature map manipulation for semantic part editing**   Manipulating feature maps of a generator directly influences the resulting synthesized image, and the clustering results of these feature maps yield semantic part representations [4]. Prior works have leveraged these characteristics of feature maps to conduct local editing tasks such as enlarging the eyes or changing the shape of ears without manual annotations [20]. However, these methods are inconvenient when it reflect the intention of user during the image editing process, presenting a considerable challenge. To address this, we propose a novel approach that transforms the feature maps into the shape of a given mask, allowing more intuitive and precise image generation. This approach significantly enhances the user experience by providing an efficient and convenient means of generating images that closely align with the user's intention.

**Mapping between feature maps and semantics**   The utility of feature maps extends beyond image generation to tasks such as few-shot semantic segmentation, offering rich semantic information from a pretrained generator[15, 45, 38, 29, 44, 2]. Techniques such as linear transformation or a simple CNN network can generate high-quality annotations from a minimal set of label-image pairs[38, 29]. While the studies mentioned above utilize feature maps to generate segmentation maps, our research goal takes a reverse approach. Inspired by [4], ours create proxy masks based on the information provided by segmentation maps or other conditions so that the model can rearrange feature maps based on the transferred clusters. Notably, LinearGAN[38] excels in enabling few-shot segmentation and achieving semantic image synthesis by finding the optimized latent codes. However, optimization in the latent space in LinearGAN lacks detailed pixel-level control and makes it challenging to apply user-desired styles properly. Our method conducts semantic image synthesis at the feature map level, significantly enhancing pixel-level control and allowing users to apply various styles from references in their preferred manner.

**Exemplar guided image synthesis**   Utilizing exemplar images rather than varying domain conditions could provide a broader controllability range. Most exemplar image generation problems aim to stylize the given condition with the style of the exemplar image [43, 46, 16, 9]. However, we focus more on reflecting the shape of the exemplar image. Techniques such as SNI[1], DAT[13], TransEditor[39], and CoordGAN[19] use dual latent spaces for image editing, which helps to disentangle the structure and style of an image. By incorporating an encoder to extract the structural latent codes of the exemplar image, it is possible to generate images that retain the shape of the exemplar while exhibiting a range of stylistic variations. However, these approaches predominantly rely on latent codes, making it challenging to replicate the structure of exemplar images precisely. While similarly employing latent codes, our method enhances spatial manipulation accuracy by rearranging the feature maps.

**Foundation model for segmentation**   The Segment Anything Model (SAM)[12] is a foundational vision model trained on over 1 billion masks. With SAM, zero-shot segmentation for general images becomes possible. The method allows the creation of segmentation masks for entire image datasets, which can then be used to train conventional SIS models. However, SAM does not cover every domain-specific semantics. Additionally, utilizing previous SIS models with SAM still requires the time-consuming process of generating mask pairs and training SIS models from scratch. Our method addresses situations where foundational models like SAM may not be suitable. It can be applied to user-specific custom datasets, offering a more efficient and adaptable solution.

## 3   Method

In this section, we explain our problem formulation and its components. Figure 1 compares the conventional semantic image synthesis and our method. The conventional semantic image synthesis methods require pixel-wise semantic label maps for all images. Conversely, our goal is to provide

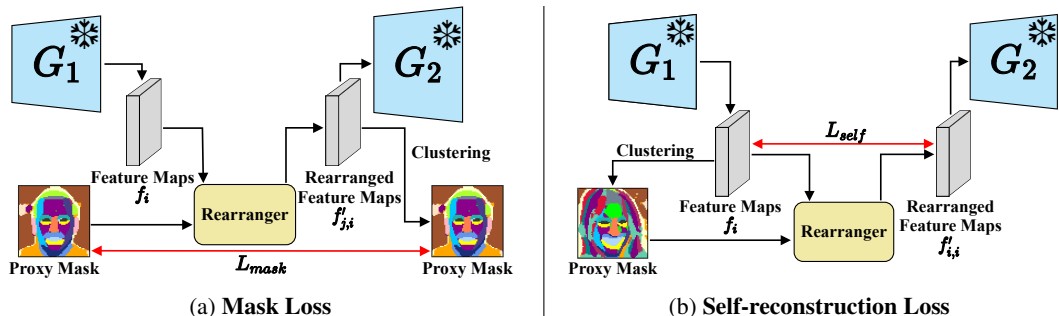

(a) **Mask Loss**     (b) **Self-reconstruction Loss**

Figure 2: **Rearranger** achieves self-supervised learning by incorporating self-reconstruction loss and mask loss, reducing the reliance on a large image-mask dataset. $G_1$ is earlier layers of the generator and $G_2$ is later layers after the proxy mask resolution.

spatial control of a pretrained unconditional generator using an input mask. We formulate the problem as rearranging intermediate features in the generator according to the input mask. As the means for providing spatial control, we introduce a proxy mask, which is defined by unsupervised clustering of the features. Then, we train a rearranger for rearranging the features according to the proxy mask. We introduced a semantic mapper that maps the input condition to the proxy mask to bridge the perception gap between the proxy mask and the input mask.

## 3.1   Self-supervised learning of spatial control

Our primary objective is to spatially rearrange the feature maps in the unconditional generator to generate appropriate images according to the specific conditions. To this end, we introduce a proxy mask that specifies the target arrangement and a rearranger, a network that utilizes an attention mechanism. Since attention is a weighted sum of values based on the similarity of the query and the key, it has proven effective for condition-based input modifications to rearrange the feature maps according to the given proxy mask.

Given a proxy mask $\mathbf{m}_i$ and original feature maps $\mathbf{f}_i = G_1(\mathbf{z}_i)$ where $\mathbf{z}$ is a random latent code, i denotes an index of the sample and $G_1$ is the front part of the model up to and including the layer corresponding to the feature maps used to generate the proxy mask, a rearranger produces feature maps through a cross-attention mechanism: $\text{Attention}(Q, K, V) = \text{softmax}\left(\frac{QK^T}{\sqrt{d}}\right)V$. Here, $Q$ denotes the embedding of the proxy mask $\mathbf{m}_i$ which is a non-linear transformation of the proxy mask., $K$ and $V$ denotes the embedding of the original feature maps $\mathbf{f}_i$, $d$ denotes the dimension of the embedding space. Formally,

$$Q = W_Q \cdot \mathbf{m}_i, \quad K = W_K \cdot \mathbf{f}_j, \quad V = W_V \cdot \mathbf{f}_j, \tag{1}$$

where $W_{(\cdot)} \in \mathbb{R}^{d \times d_{\mathbf{f}}}$ are learnable embedding matrices [8, 30]. For simplicity, we use rearranged feature maps as $\mathbf{f}'_{i,j} = \text{Attention}(\mathbf{m}_i, \mathbf{f}_j, \mathbf{f}_j)$ instead of $\text{Attention}(Q, K, V)$. With this attention mechanism, the image rendering can be represented as $\mathbf{X}_{\text{fake}} = G_2(\mathbf{f}'_{i,j})$, where $(i, j)$ is a random pair of latent codes, $\mathbf{X}_{\text{fake}}$ denotes the synthesized image and $G_2$ is the rest part of generator consists of later layers of generator. Further, we define proxy mask $\mathbf{m}_i$ in terms of $\mathbf{f}_i$ via Clustering which is a K-means clustering algorithm , expressed as $\mathbf{m}_i = \text{Clustering}(\mathbf{f}_i)$.

As shown in Figure 2, to make output of rearranger more accurately reflect the mask given as condition, we encourage the network to reconstruct the proxy mask from the rearranged feature maps by

$$\mathcal{L}_{\text{mask}} = \text{CrossEntropy}(\mathbf{m}'_{i,j}, \mathbf{m}_i) \tag{2}$$

Where $\mathbf{m}'_{i,j} = \text{Clustering}(\mathbf{f}'_{i,j})$ denotes the proxy mask of rearranged feature maps.

However, to maintain the style of source latent, we encourage the network to self-reconstruct when a proxy mask created from the same latent, is used as the query for attention:

$$\mathcal{L}_{\text{self}} = \sum_i \left\| \mathbf{f}'_{\text{i,i}} - \mathbf{f}_i \right\|_2 \qquad (3)$$

Our overall learning pipeline operates as follows. We first prepare the original feature maps $\mathbf{f}_i$ from a random latent code $\mathbf{z}_i$. We then apply K-means clustering on the feature maps to derive segmentation-like masks. We apply the loss as mentioned above functions using these masks and feature maps, which are created from different latents. These processes work in a self-supervised manner, enabling the rearranger to learn effectively and autonomously by leveraging the representation of the pretrained generator and an unannotated dataset.

Finally, the overall training term, denoted as $\mathcal{L}_{\text{total}}$, is composed of the non-saturating adversarial loss with R1-regularization [5, 18], self-reconstruction loss, and mask reconstruction loss.

$$\mathcal{L}_{\text{total}} = \mathcal{L}_{\text{adv}} + \lambda_{\text{self}}\mathcal{L}_{\text{self}} + \lambda_{\text{mask}}\mathcal{L}_{\text{mask}} + \lambda_{\text{R1}}\mathcal{L}_{\text{R1}} \qquad (4)$$

### 3.2 Semantic guide

In the earlier sections, we have introduced an approach to provide spatial control for a pretrained unconditional GAN through proxy masks generated via unsupervised feature maps clustering. However, the process lacks a direct connection from the input mask to the proxy mask. To bridge this gap, we propose a semantic mapper to convert the user given condition including input mask into a proxy mask, as illustrated in Figure 3. This conversion can be represented as Mapper($\mathbf{c}$), where $\mathbf{c}$ is the input condition.

We begin the process by generating images and their corresponding proxy masks from random latent codes. Next, we employ a one-shot segmentation network, as suggested in [29], to prepare a set of mask and proxy mask pairs. These

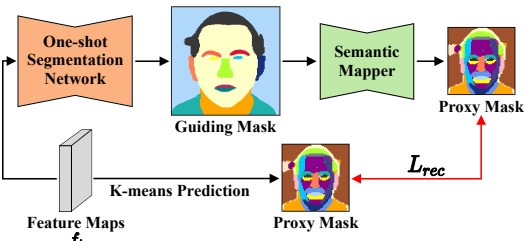

Figure 3: **Semantic Mapper** transforms input masks that the generator cannot comprehend into proxy masks, enabling the generator to understand and synthesize the corresponding output. When training the semantic mapper, a one-shot segmentation network and reconstruction loss are used.

pairs are used to train our semantic mapper, allowing it to generate a proxy mask $\mathbf{m}_i$ from the input mask $\mathbf{m}'_i = \text{SegNet}(\bar{\mathbf{f}}_i)$ by

$$\mathcal{L}_{\text{rec}} = \text{CrossEntropy}(\mathbf{m}'_i, \mathbf{m}_i) \qquad (5)$$

Here, SegNet denotes the one-shot segmentation network and $\bar{\mathbf{f}}_i$ is the concatenation of several feature maps for the SegNet input. Through this training, we establish a connection that bridges the perception gap between human and the generator, enabling the generator to better understand and integrate the semantic content intended by the user.

Furthermore, we have also incorporated an adversarial training strategy in the semantic mapper's training process. We utilize a discriminator identical to the one used in StyleGAN2 [11], which discriminates between real and fake images generated with proxy masks. The overall training term for the semantic mapper comprises adversarial loss, reconstruction loss, and R1 regularization.

$$\mathcal{L}_{\text{total}} = \lambda_{\text{adv}}\mathcal{L}_{\text{adv}} + \lambda_{\text{rec}}\mathcal{L}_{\text{rec}} + \lambda_{\text{R1}}\mathcal{L}_{\text{R1}} \qquad (6)$$

Through this Semantic Guide, we have enabled our generator to capture and integrate the desired semantic information effectively. Consequently, the synthesized images produced by our method closely align with the user's intended semantic content.

## 4 Experiments

We have conducted a range of experiments to highlight the adaptability of our approach across diverse datasets, including CelebAMask-HQ [14], LSUN Church, LSUN Bedroom [41], FFHQ [10],

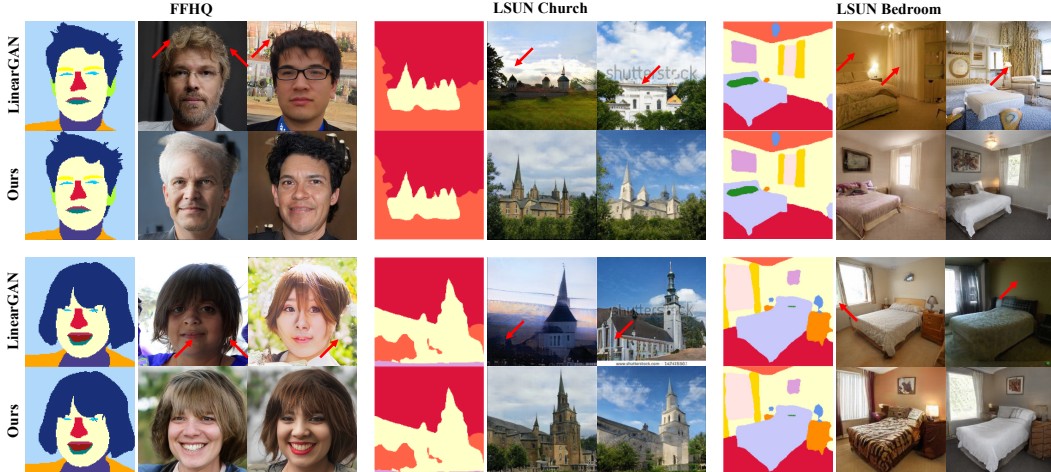

Figure 4: Comparison with LinearGAN when receiving the same mask. LinearGAN is an indirectly optimized method that adjusts the mask through optimization, while our approach is designed to create images that better fit the given mask. Our method performs better by producing more accurate images that align with the masks. Both LinearGAN and Ours use a single mask to train each model.

and AFHQ [3]. Moreover, we used these datasets for conditional image synthesis under various conditions, including segmentation masks, sketches, and user scribbles. Regarding model architecture, we employed the generator and discriminator of StyleGAN2 [11] as a pretrained model. Furthermore, we used an architecture similar to SPADE [21] and a simple UNet architecture as a mapper architecture for segmentation masks and different conditions, respectively. For more detailed information about these experiments and settings, please refer to the Appendix. In addition, we include an ablation study in the Appendix that shows the validity of our proposed loss function and the appropriate number of clusterings.

## 4.1 Semantic image synthesis

### 4.1.1 Comparison to few-shot semantic image synthesis method

We present a comparison with LinearGAN [38], a method specifically tailored for few-shot semantic image synthesis utilizing a pretrained generator. LinearGAN optimizes the latent vector, which most closely matches the shape of the mask, using few-shot semantic segmentation for semantic image synthesis. Our method surpasses LinearGAN in semantic image synthesis performance with only a single imput mask. To provide an accurate comparison with LinearGAN, we also conducted comparisons in a few-shot setting.

Figure 4 illustrates the difficulty of LinearGAN in accurately reflecting the overall mask structure. While LinearGAN can generate images that align with the input mask for datasets with less shape diversity, such as human faces, it struggles with LSUN church or LSUN bedroom because of the diverse shape of the datasets. This challenge arises as LinearGAN often finds it difficult to identify a latent closely resembling the shape. Consequently, it fails to reflect intricate shapes effectively. In contrast, our method adjusts the feature maps to correspond with the shape of the input mask, demonstrating better results with detail.

The quantitative comparison, as seen in Table 1, utilizes the mIoU between the input mask and the mask generated by a pretrained segmentation network. This metric evaluates the accuracy of the model in reflecting the mask. Our method, surpassing LinearGAN across all datasets, shows its precision in reflecting the input mask, even with limited data. Remarkably, this superiority is maintained irrespective of the number of input masks. This advantage is both qualitatively and quantitatively evident, validating the effectiveness of our approach in few-shot semantic image synthesis.

| N | Method | mIoU | | |
|---|---|---|---|---|
| | | Church | Bedroom | FFHQ |
| 1 | LinearGAN | 16 ± 1.4 | 17.5 ± 2.0 | 37.2 ± 0.8 |
| | Ours | **23.8 ± 4.4** | **35.6 ± 4.0** | **49.8 ± 1.0** |
| 4 | LinearGAN | 18 ± 1.3 | 21.6 ± 0.9 | 39.1 ± 0.5 |
| | Ours | **27.6 ± 2.09** | **44.3 ± 2.4** | **54.3 ± 1.18** |
| 8 | LinearGAN | 19.6 ± 0.5 | 21.7 ± 0.8 | 39.4 ± 0.9 |
| | Ours | **28.7 ± 1.23** | **44.7 ± 3.0** | **54.4 ± 1.7** |
| 16 | LinearGAN | 20.4 ± 0.6 | 22.3 ± 0.4 | 40 ± 0.2 |
| | Ours | **28.8 ± 1.08** | **45.7 ± 0.9** | **56.7 ± 1.19** |

Table 1: mIoU results of LinearGAN and Ours on various datasets. N means number of mask images used in few-shot setting.

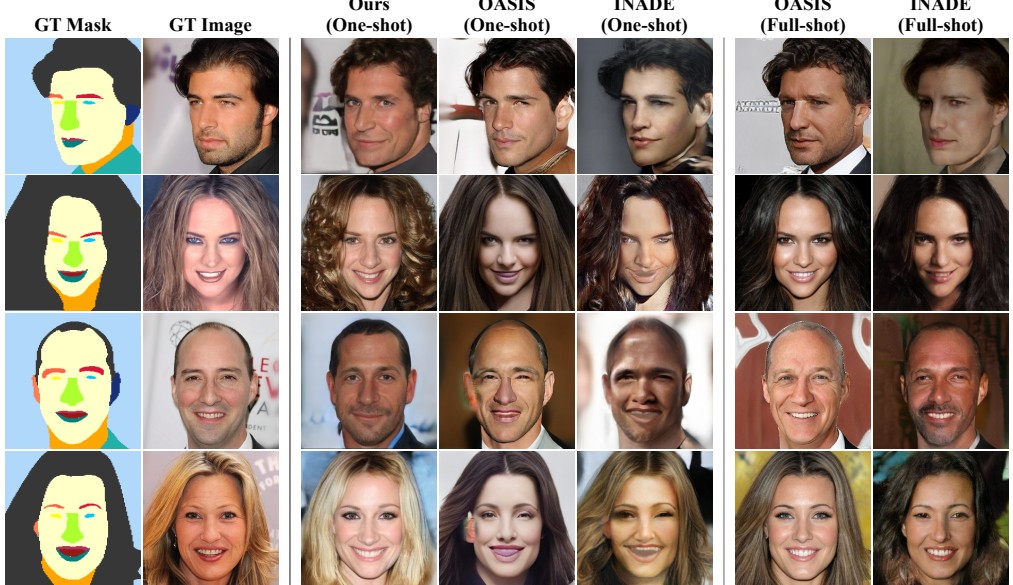

Figure 5: The comparison results between our method and supervised models using the same mask (Including one-shot setting). Compared to existing SIS methods, Ours produces results that are closer to the ground truth. Additionally, Our result shows the most natural-looking output.

#### 4.1.2 Comparison to full-shot method

Furthermore, we carried out both quantitative and qualitative assessments of our proposed method and the baseline approaches with CelebAMask-HQ [14], LSUN Church, LSUN Bedroom [41], and FFHQ [10].

As illustrated in Figure 5, we adopted the RepurposeGAN [29] approach to generate one-shot data pairs for comparison under a one-shot setting. While the results from competitors frequently display unnatural artifacts, our method yields more natural and closer-to-ground truth outcomes. Also, as depicted in Table 2, our method has a higher mIoU than existing techniques with one-shot training data pairs. The reason is that our approach excels in accurately segmenting large mask regions. Our method shows lower mIoU than the conventional models trained with full-shot 28K image-annotation pairs because IoU decreases for fine-grained classes that are not included in the one-shot setting, such as accessories with low frequencies. However, when it comes to the FID, our work significantly outperforms the one-shot settings. It is typically superior to the full-shot methods, even with just a single annotation. In addition, we have included results for other metrics, such as LPIPS and Precision and Recall, in the Appendix.

|        | FID  | mIoU(vs gt mask) | Acc(vs gt mask) |
|--------|------|------------------|-----------------|
| SEAN   | 28.1 | 75.9             | 92.3            |
| OASIS  | 20.2 | 74.0             | 91.5            |
| INADE  | 21.5 | 74.1             | 93.2            |
| SDM[34]| 18.8 | 77.0             | n/a             |
| OASIS (1-shot) | 25.5 | 45.5     | 82.7            |
| INADE (1-shot) | 28.1 | 44.3     | 83.8            |
| Ours (1-shot)  | 18.5 | 53.1     | 88.2            |

Table 2: Quantitative comparison with SEAN, OASIS(1-shot / full-shot), INADE(1-shot / full-shot), SDM. mIOU and Acc are measured by comparing with the ground truth mask.

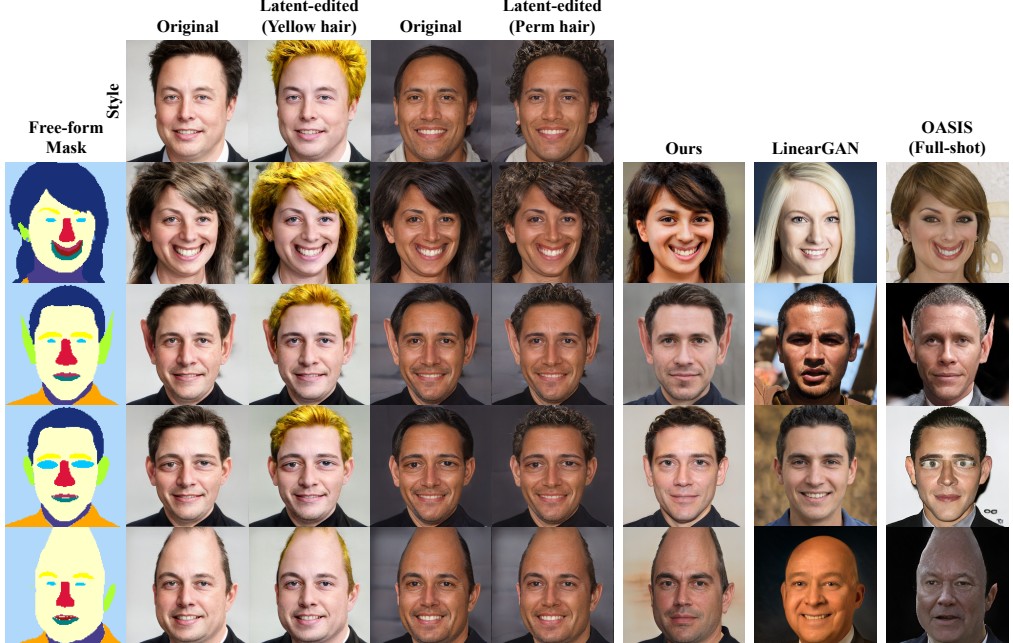

Figure 6: Result of image synthesis with free-form mask. Images in the first row are inverted images in StyleGAN and their edited images with the HairCLIP [36]. Images are converted to the spatial shapes of both masks and images in the first column while preserving styles and identities. We also show a comparison with LinearGAN [38] and OASIS [25] in the last three columns. For comparison we sample the images with random noise for all models

## 4.2 Free-form image manipulation

In this subsection, we show the flexibility of our proposed method in facilitating free-form image manipulation. Our approach underscores its capability of handling masks that deviate substantially from typical training distributions, such as oversized human ears and big oval eyes. Figure 6 illustrates the image synthesis results when such unconventional masks are used as inputs. Even in cases where the masks deviated from the typical human form, our method successfully synthesized results that corresponded appropriately to the provided masks. In contrast, when LinearGAN [38] and OASIS [25] receive a mask that falls outside the typical human range as a shape guide, they fail to generate results that align with the shape indicated by the mask.

Moreover, the versatility of our method is further emphasized by its compatibility with latent editing techniques such as HairCLIP [36] as shown in Figure 6. While latent editing changes the hairstyle to a yellow color or perm style, our method preserves the shape of images corresponding to given masks. See the Appendix G for additional samples of generated images conditioned with free-form masks.

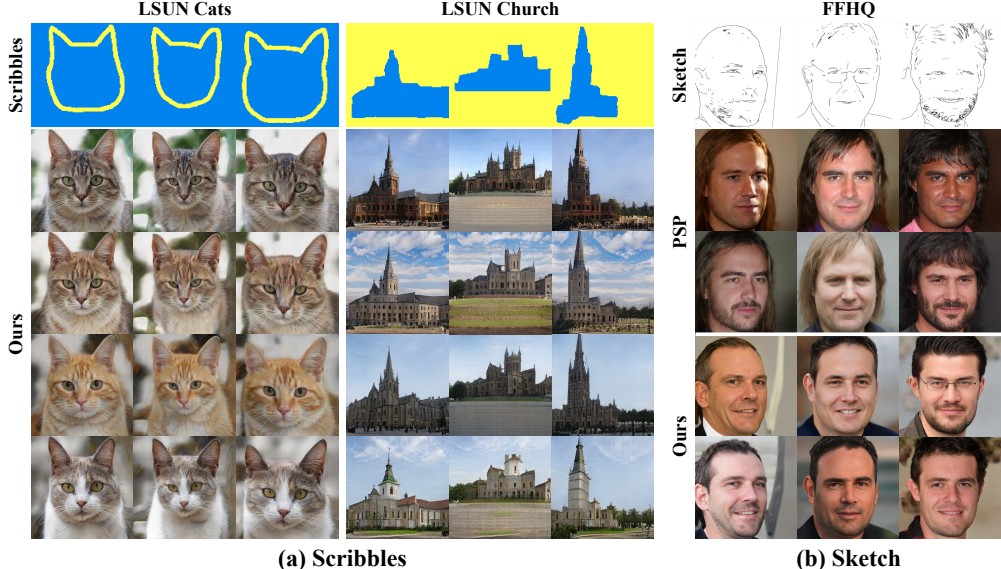

**(a) Scribbles**  **(b) Sketch**

Figure 7: Image synthesis results when diverse types of input masks are entered, such as scribbles and sketches.

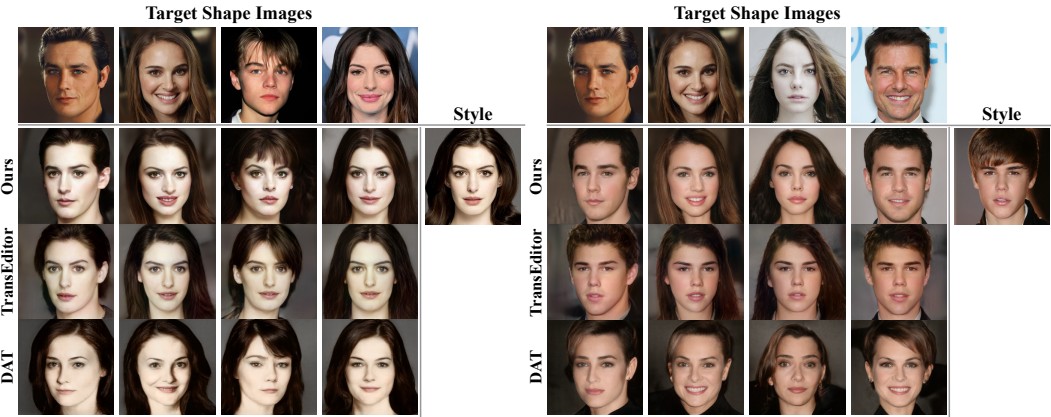

Figure 8: Image generation results with a real image as a guide. The top row consists of images that serve as the style input, while the left column contains target image that determine the desired shape.

## 4.3 Conditional image synthesis using a variety of conditions.

This part broadens the scope of semantic image synthesis to include just segmentation masks and diverse conditions like sketches and user scribbles. Figure 7(a) demonstrates the outcome when simple scribbles are used as an input mask. Remarkably, even with a scribble mask created with just a single line, our method can generate image synthesis that is well-suited to the indicated shape. Moreover, this user-friendly aspect further enhances the usability of our approach. Additionally, we visualize our results and compare them qualitatively with pSp [23], a GAN inversion approach demonstrating its ability to invert various inputs into latent codes, not solely real images. As illustrated in Figure 7(b), our comparison with pSp, especially in semantic image synthesis using sketch inputs, reveals a significant difference. pSp can generate coarse structures effectively with pSp encoder and style mixing, but it struggles to capture and reproduce the fine details of the input conditions. On the other hand, our method surpasses pSp with more corresponding results with fewer restrictions of matching latent code because of the flexible spatial control of our model using feature maps rearranging. In the Appendix E, we further demonstrate the proficiency of our method in generating images based on various input conditions with implementation details. This additional evidence underscores the versatility of our method and superior performance in conditional image synthesis.

### 4.4 Exemplar-guided image generation

Beyond the specified conditions, our model produces an output image that has a similar structure to a provided sample. The results of image generation are displayed in Figure 8. The latent codes are derived from real images by utilizing the e4e inversion technique [28] that was trained with the CelebAMask-HQ training set. This inverted latent code enables us to obtain a proxy mask of the sample, thereby aligning the generated images with the real image structure.

In Figure 8, it is evident that our generated results better preserve shape in comparison to other exemplar-guided image generations. The results of Transeditor [39] and DAT [13] do not preserve the original shape as effectively, with DAT even showing a disparity in style. Although competitors use dual latent spaces to portray exemplar structure more accurately, the structural latent code does not always reflect the shape correctly. In contrast, our method more accurately mirrors the exemplar shape because the Rearranger directly forms the feature maps with the proxy mask.

## 5 Conclusion

Semantic image synthesis has received significant attention for its ability to reflect intentions on images at a very detailed level, but achieving this requires substantial data. Therefore, we proposed our method as a solution to address this issue. Our method rearranges the features of a pretrained generator to enable detailed control over images, and all elements of our approach are self-supervised, except for a single annotation mapped to human perception. With just one annotation, we achieve precise image control, as demonstrated through experiments in semantic image synthesis with various input conditions. Our method also showed data efficiency in situations where the pixel-level annotation changes. Overall, our approach demonstrated a high level of detail control without the dependence on annotations.

**Limitations** Our one-shot results outperform other competitors, but a slight performance gap remains when compared to full-shot outcomes. Therefore, future work will make it possible to outperform to beat full-shot performance. Also, our approach operates at the feature map level, which, while enabling the natural incorporation of masks, needs to achieve complete pixel-level control. For more details, please refer to the Appendix D.

## Acknowledgement

This work was supported by the National Research Foundation of Korea(NRF) grant (No. 2022R1F1A107624111) funded by the Korea government (MSIT).

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
