# Supplementary Material for
# Semantic Image Synthesis with Unconditional Generator

**JungWoo Chae**[12*] **Hyunin Cho**[1*] **Sooyeon Go**[1]   **Kyungmook Choi**[1]    **Youngjung Uh**[1†]

[1]Yonsei Unviersity, Seoul, South Korea
[2]LG CNS AI Research, Seoul, South Korea
{cjwyonsei, hyunin9528, sooyeon8658, kyungmook.choi, yj.uh}@yonsei.ac.kr
{cjwoolgcns}@lgcns.com

## A   Implementation details

### A.1   Architectural designs

We provide additional details of the architectural designs for our rearranging network and semantic mapper. For the rearranging network, we incorporate a single head cross-attention module, which is surrounded by two 4-layer residual blocks on either side. In Figure S1, the cross-attention module operates by computing an attention matrix from the query (the proxy mask) and the key (feature maps). This process enables the value (feature maps) to be rearranged (through a weighted sum) to align with the form of the query, thereby reflecting their strong correspondence. To reflect dissimilarity between different pixels in a mask, we add a sinusoidal position encoding before the residual block that precedes the module.

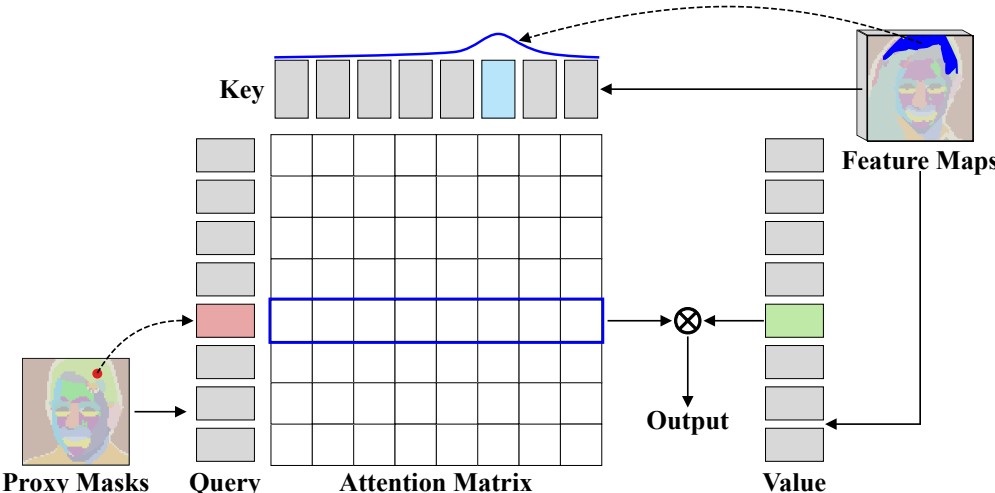

Figure S1: **Illustration of the cross-attention operation.** The module computes an attention matrix using the query (proxy mask) and key (feature maps), enabling the rearrangement of the value (feature maps) according to the shape of the query.

---

*Equal contribution.
†Corresponding author

37th Conference on Neural Information Processing Systems (NeurIPS 2023).

For the semantic mapper, we adopt the architecture of OASIS [9], which takes the input condition and generates a $64^2$ resolution output proxy mask, except input noise. This method ensures a high level of incorporation of the input mask in the proxy mask. The input noise is removed because its stochasticity slows down the training. However, when the input condition is not a semantic segmentation mask, it is not appropriate to use a SPADE layer [7]. As a solution, we added a U-Net structured decoder to the encoder which has same structure with discriminator of the StyleGAN2 [4], allowing the shape of the input mask to be preserved all the way to the output proxy mask.

## A.2 Training details

In our training setup, we follow a similar approach to that of StyleGAN2[4], where the learning rate is set to 0.002 for both the generator and discriminator. Additionally, the weight for $R_1$ regularization is set to 10.

We prepare the proxy masks from the centroids obbtained by clustering feature maps of 256 sample images. The size of the feature maps is $64^2$. To achieve this, we employ K-means clustering [2]. Given the need for balancing between high correspondence and image quality, we empirically set the weights of our loss terms. Specifically, we set $\lambda_{mask}$ to 1.0 and $\lambda_{self}$ to 10.0. To prevent shortcuts of self-reconstruction loss, we apply a random horizontal flip augmentation. Furthermore, we alternate training with a proxy mask generated from the same noise as the feature maps used for self-reconstruction and a proxy mask generated from random noise in each iteration.

The training for our rearrangement network was conducted using the CelebAMask-HQ [5], LSUN Church, and LSUN Bedroom datasets [13], each at a resolution of 256. We adopted a strategy where $\mathcal{L}_{self}$ and $\mathcal{L}_{mask}$ were computed for every iteration during the initial 100k iterations, whereas $\mathcal{L}_{adv}$ was computed once every five iterations. Subsequently, all three loss components were computed for each iteration over the next 40k iterations. When training with the 1024 resolution FFHQ [3] and 512 resolution AFHQ[1] datasets, we computed $\mathcal{L}_{self}$ and $\mathcal{L}_{mask}$ at each iteration for the first 150k iterations, with $\mathcal{L}_{adv}$ being evaluated once every 5 iterations. Subsequently, all three loss terms were computed for each iteration over the following 65k iterations.

For the training of the semantic mapper, we initially trained for the first 100k iterations using only $\mathcal{L}_{recon}$ with $\lambda_{self}$ to 10.0. Then, for the subsequent 10k iterations, $\mathcal{L}_{adv}$ was added to calibrate the generated image within its domain, during which we adjusted $\lambda_{recon}$ to 100.0. It should be noted that all $\lambda_{adv}$ values used in the mapper and rearranging network were consistently set to 1.0.

The entire training completes within a day for data at a resolution of 256 using an NVIDIA RTX 3090 GPU. For higher resolution configuration ($1024^2$), the training completes within two days. This efficient training process highlights the feasibility of our approach for practical applications.

## B Ablation study

### B.1 Proposed loss function

To demonstrate the influence of the additional losses introduced in our method, we provide both quantitative and qualitative ablations in Figure S2 and S3, respectively. We conducted these experiments on the CelebA dataset, comparing three scenarios: using only $\mathcal{L}_{adv}$, adding $\mathcal{L}_{self}$, and using all three losses, including $\mathcal{L}_{mask}$. The Fréchet Inception Distance (FID) is measured at each iteration, and the degree of attribute preservation is demonstrated through a curation of the images produced at the final convergence point. When evaluating the FID , we generated a dataset consisting of 28,000 images by employing two latent codes. One latent code was used for generating proxy masks, while the other code was utilized for defining the desired style. These generated images were then compared with a dataset of 28,000 training images for FID calculation.

As evident from the plot in Figure S2, the presence or absence of $\mathcal{L}_{self}$ significantly affects the speed of convergence. This is because the self mask serves as a guide for the rearranging network, particularly in the early stages of training. However, if only $\mathcal{L}_{self}$ is incorporated, the network tends to disregard the proxy mask, favoring the restoration of its original state, as illustrated in the corresponding figure.

Thus, the inclusion of $\mathcal{L}_{mask}$ serves to counterbalance this trend, ensuring that the proxy mask is not ignored and is effectively utilized in the image generation process. These results demonstrate the

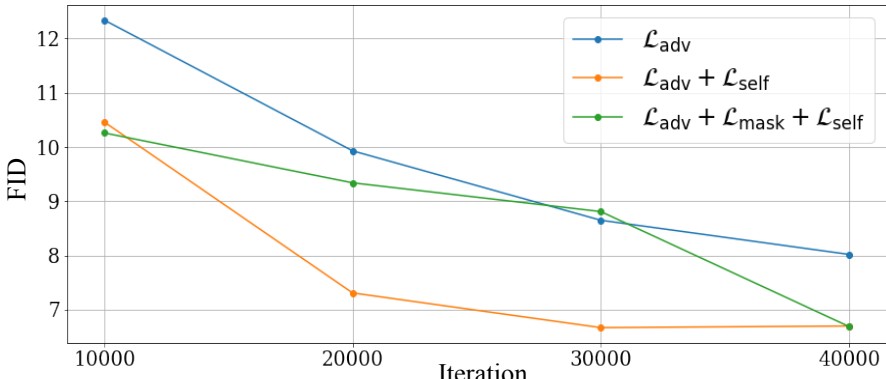

Figure S2: **Quantitative studies to demonstrate the impact of additional losses introduced in our method** $\mathcal{L}_{\text{self}}$ leads to a rapid decrease in FID, thus demonstrating its contribution to achieving fast convergence.

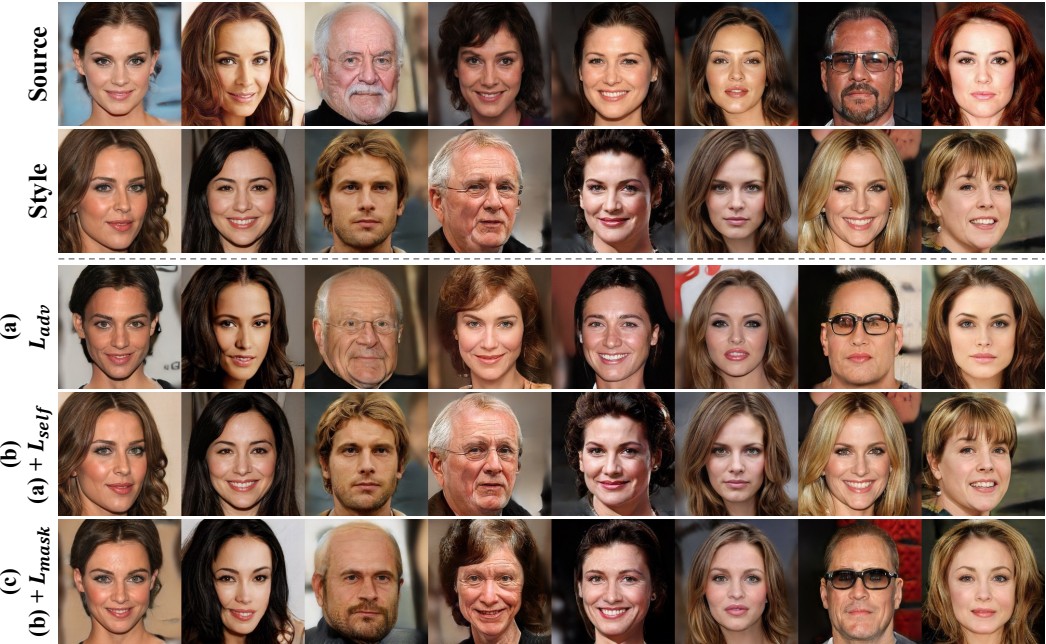

Figure S3: **Qualitative studies to demonstrate the impact of additional losses introduced in our method** (a) $\mathcal{L}_{\text{loss}}$ contributes to producing realistic results. (b)$\mathcal{L}_{\text{self}}$ increases the preservation rate of style attributes but tends to ignore the shape of source image. (c)$\mathcal{L}_{\text{mask}}$ effectively reflects the shape of the source image.

critical role and effectiveness of our proposed loss components in achieving high-quality, attribute-preserving image generation.

## B.2 Number of clusters

The number of clusters significantly impacts the shape of the proxy mask and, consequently, the generated image. Therefore, we carry out a qualitative ablation study on how the number of clusters influences the generated image, the results of which are presented in Figure S4.

For this evaluation, we generate images by applying rearrangement using the feature map and the proxy mask obtained from random noise. Our results, presented in Figure S4, show that as the number

of clusters increases, detailed elements like glasses are more accurately reflected in the proxy mask. This improvement can be attributed to the increased capacity to capture finer image details as the number of classes in the proxy mask escalates.

However, when the number of clusters is low, we noticed that the influence of the position embedding becomes dominant over the class embedding, resulting in the proxy mask being overlooked and the image not transforming as intended.

Nonetheless, caution is warranted when overly increasing the number of clusters. Beyond a certain point, the proxy mask begins to diverge from the semantics understood by humans, which not only confuses the semantic mapper but also hampers generating a suitable proxy mask from the input mask. Therefore, for the convenience of our experiments, we decided to fix the number of clusters at 25.

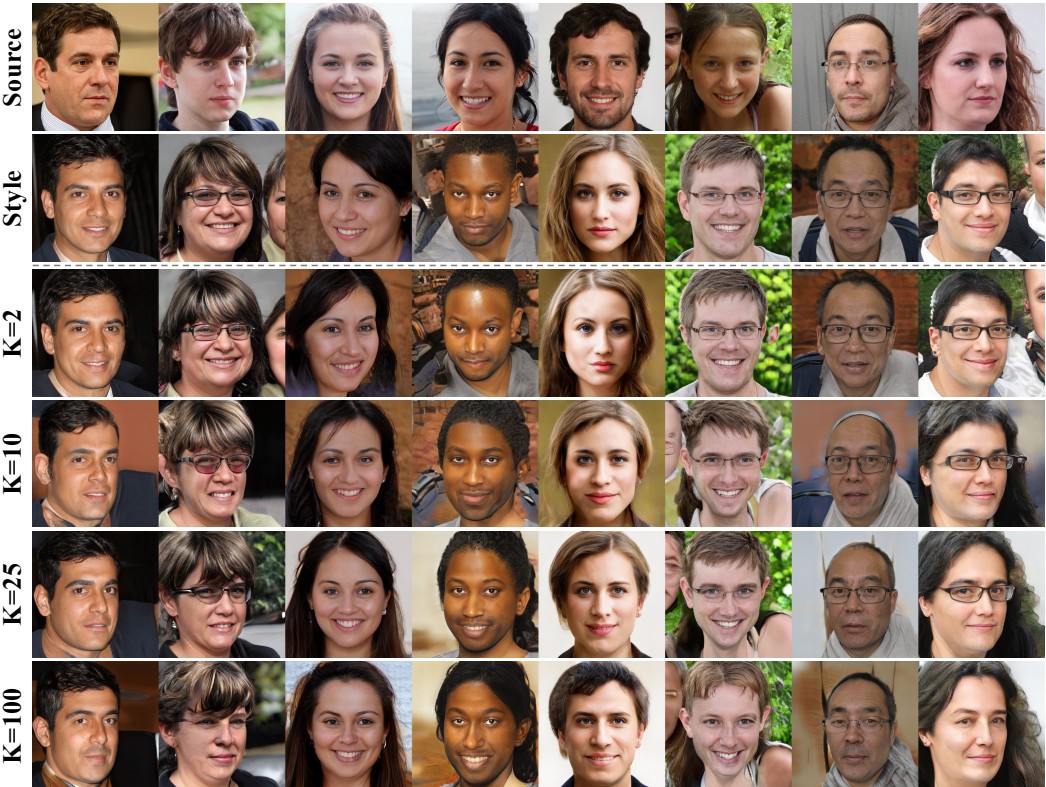

Figure S4: **Qualitative studies to demonstrate the impact of number of clusters** As the number of clusters increases, more detailed semantic parts, such as glasses, can be separated. K denotes number of clusters.

## C Additional quantitative comparison

### C.1 Additional diversity metric

Additionally, we assess result diversity using the LPIPS metric. LPIPS diversity of our method surpasses INADE and OASIS, but there is no clear winner when combining precision and recall. Table S1, summarizes these metrics. To measure LPIPS, we follow INADE[10], evaluating overall diversity by generating 10 image groups with random noise and calculating diversity scores between two groups. We compute 10 scores, averaging them to reduce fluctuation from random sampling.

### C.2 Training time comparison

To demonstrate the efficiency of our model, we evaluated our model to OASIS [9] using the FID over time. As depicted in Figure S5, training Rearranger and Semantic Mapper consecutively shows a

faster convergence speed than our competitor. Furthermore, if we have trained Rearranger, our model only requires the retraining of the Semantic Mapper when there is a change in the target semantic segmentation mask classes.

|  | LPIPS | Precision | Recall |
|---|---|---|---|
| Ours | 0.45 | 0.76 | 0.22 |
| INADE | 0.36 | 0.86 | 0.19 |
| OASIS | 0.30 | 0.69 | 0.36 |

Table S1: Additional quantitative comparison with INADE and OASIS.

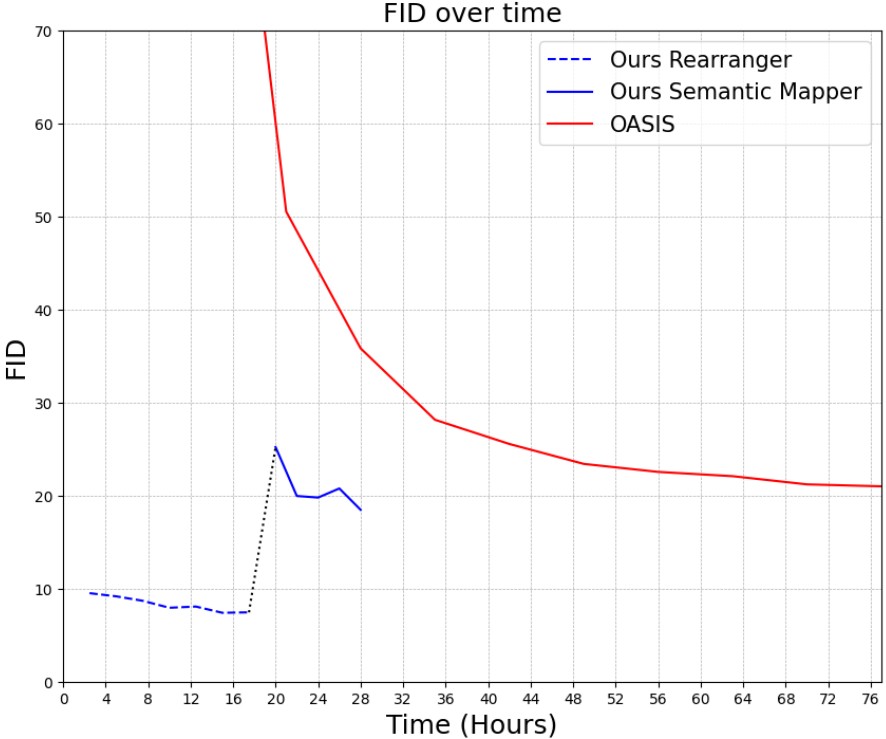

Figure S5: FID of generated images when time passes. The result shows that our work converges faster than OASIS.

## D Failure cases

The proxy mask in our method is constructed from the clustering of intermediate feature maps of the pretrained Generator, especially size of 64. Consequently, the variance in resolution between the full image and the mask often results in an imperfect fit of the image to the given mask. Figure S6 illustrates instances where our method encounters limitations due to this resolution difference. In the highlighted red boxes within the figure, it is evident that intricate features, such as the length of a mouth or individual stray hairs, are not faithfully represented in the output image. This provides an opportunity for future work focused on enhancing pixel-level manipulability.

## E Conditional image synthesis using different types of conditions

In Figure S7, S8, S9, we present randomly sampled images generated by our model with different types of input conditions, utilizing Photo-sketch [6], HED [12], and depth [8] with various datasets

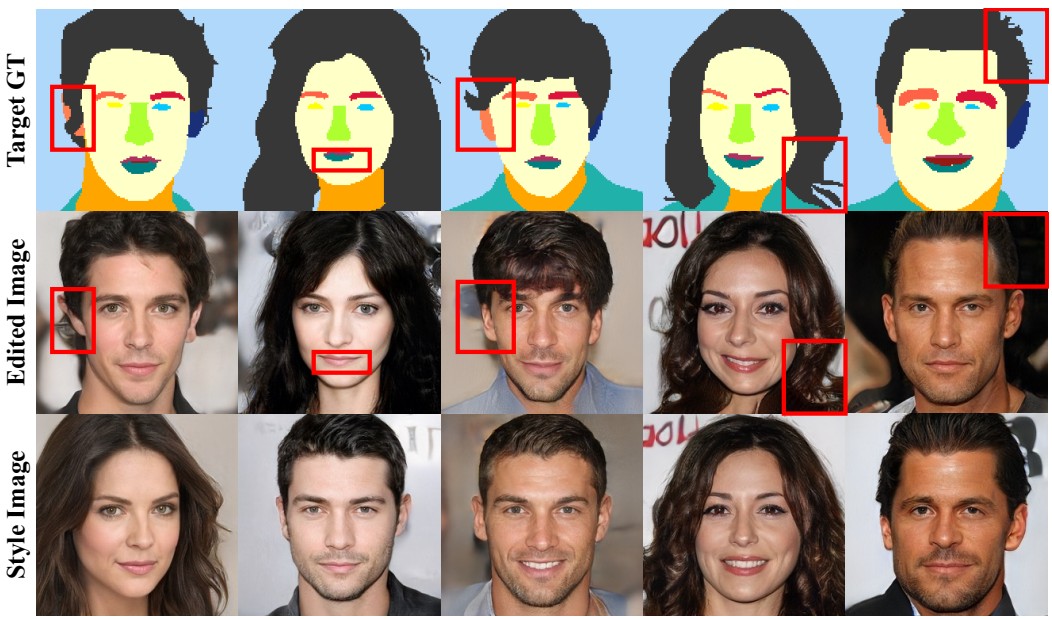

Figure S6: **Failure cases** of our method. Boxed areas show where the generated images do not match the corresponding masks.

such as FFHQ, AFHQ, LSUN Bedroom, and LSUN Church. We trained the mapper using conditions generated by passing training data through the corresponding pre-trained networks.

To generate training pairs for the mapper without relying on labor-intensive human annotations, we employed the methodology proposed by [11] for generating segmentation maps and scribbles. Additionally, for the sketch, HED, Photo-sketch, and depth scenarios, we leveraged simple individual pretrained networks or tools to acquire pairs of generated images and their respective conditions.

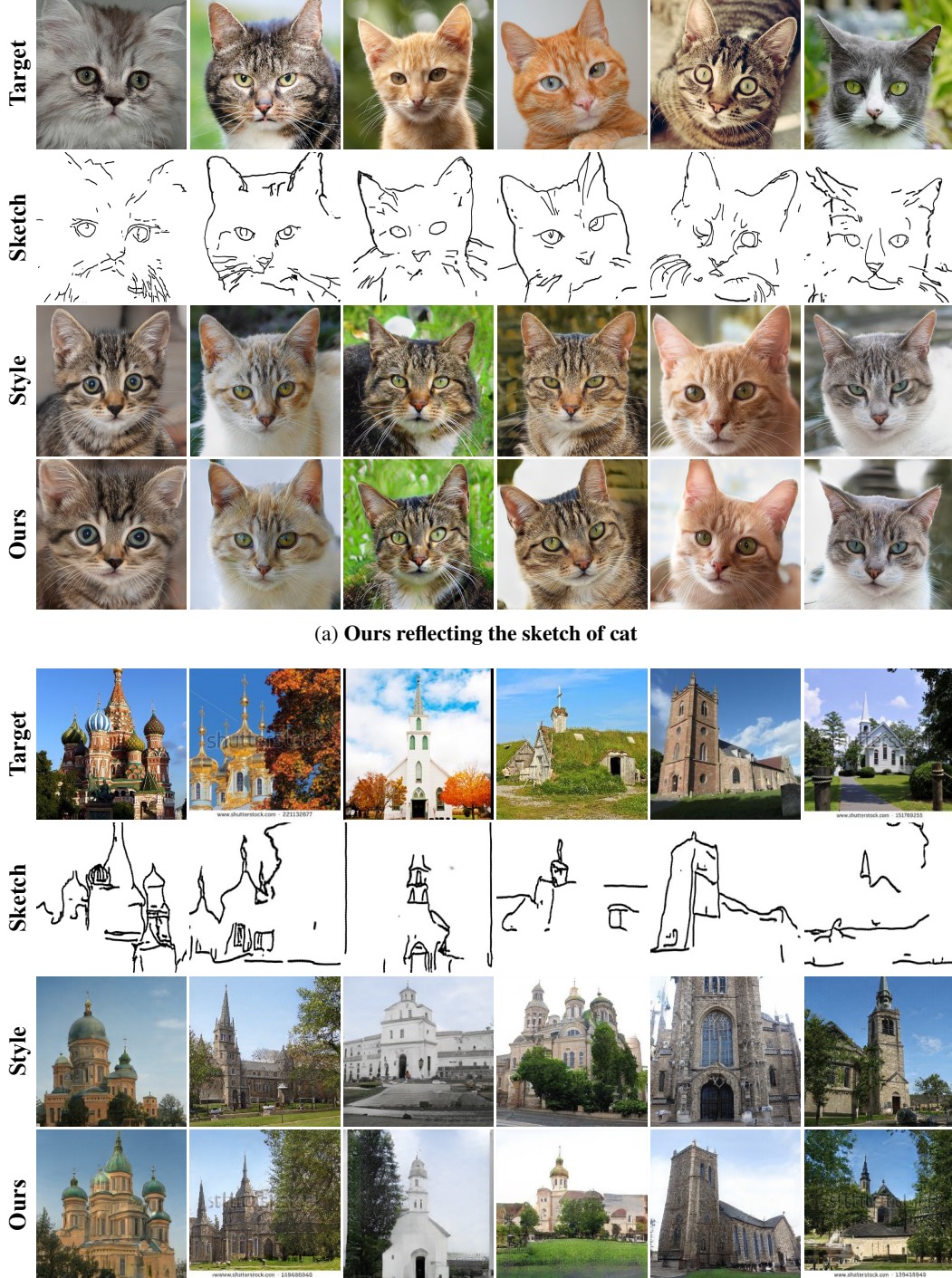

(a) **Ours reflecting the sketch of cat**

(b) **Ours reflecting the sketch of church**

Figure S7: **Ours generated using the sketch on various dataset** (a) Ours accurately captures the shape obtained from the target image. The cat's facial angle in ours is appropriately aligned with the target shape. (b) Ours is capable of creating a church position similar to the church in the target image based solely on the sketch results. In addition, the style image's church, sky, and tree colors are applied equally in ours.

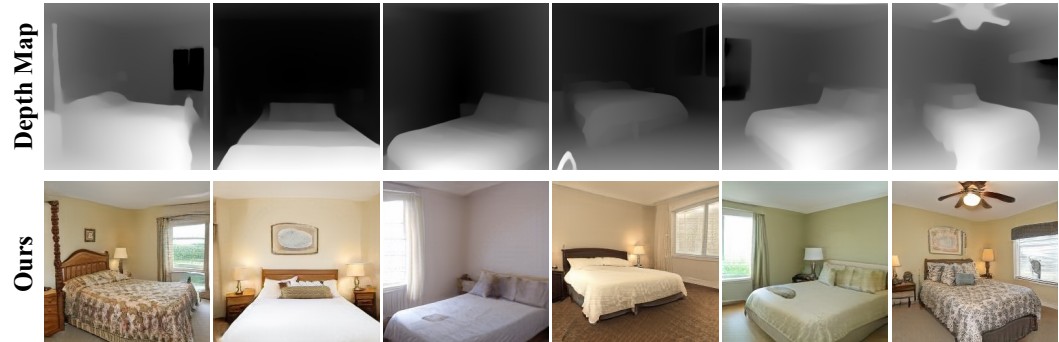

Figure S8: **Ours generated using the depth map on LSUN Bedroom** Ours effectively reflects the positions and shapes of objects such as beds or windows that can be observed in the depth map.

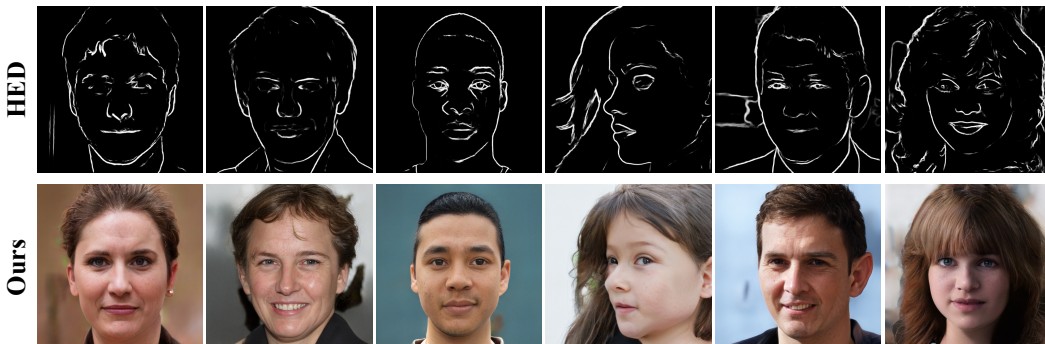

Figure S9: **Ours generated using the hed mask on FFHQ** Ours is generated to align with the facial shape determined by HED. It accurately captures aspects such as hairstyle, facial angle, and eye size in accordance with the shape outlined by HED.

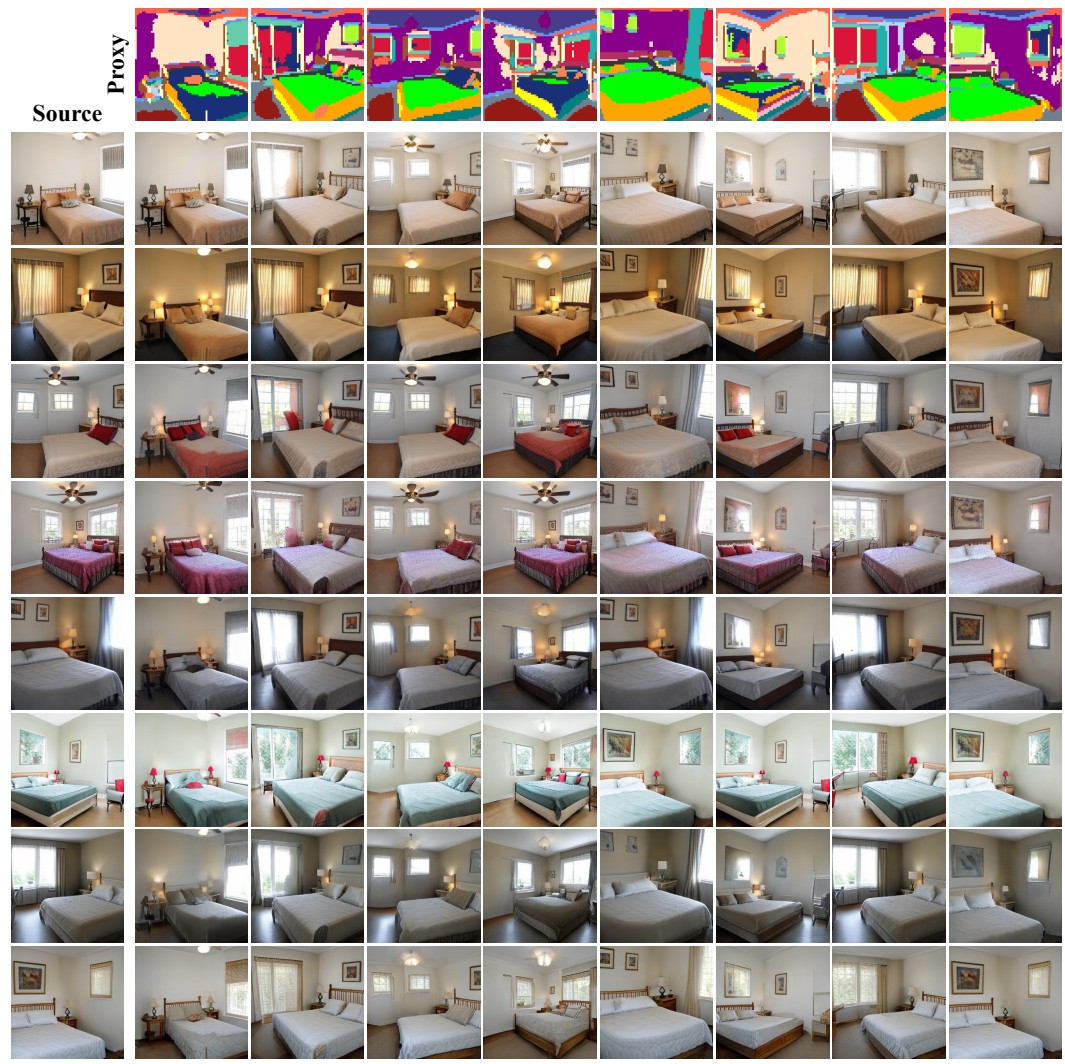

Figure S10: **Ours using the proxy mask obtained from random noise on LSUN Bedroom dataset** By generating images using the proxy mask without considering the perception gap, it is possible to create images of a higher quality.

## F    Semantic image synthesis with proxy mask

Figure S10, S11, S12, showcase the results of image generation solely using the proxy mask. Specifically, we generate images by conducting rearrangement the feature map with the proxy mask, both of which are derived from random noise. The output images illustrate the capability of our method to reflect both the attribute of the source and the shape of the target mask.

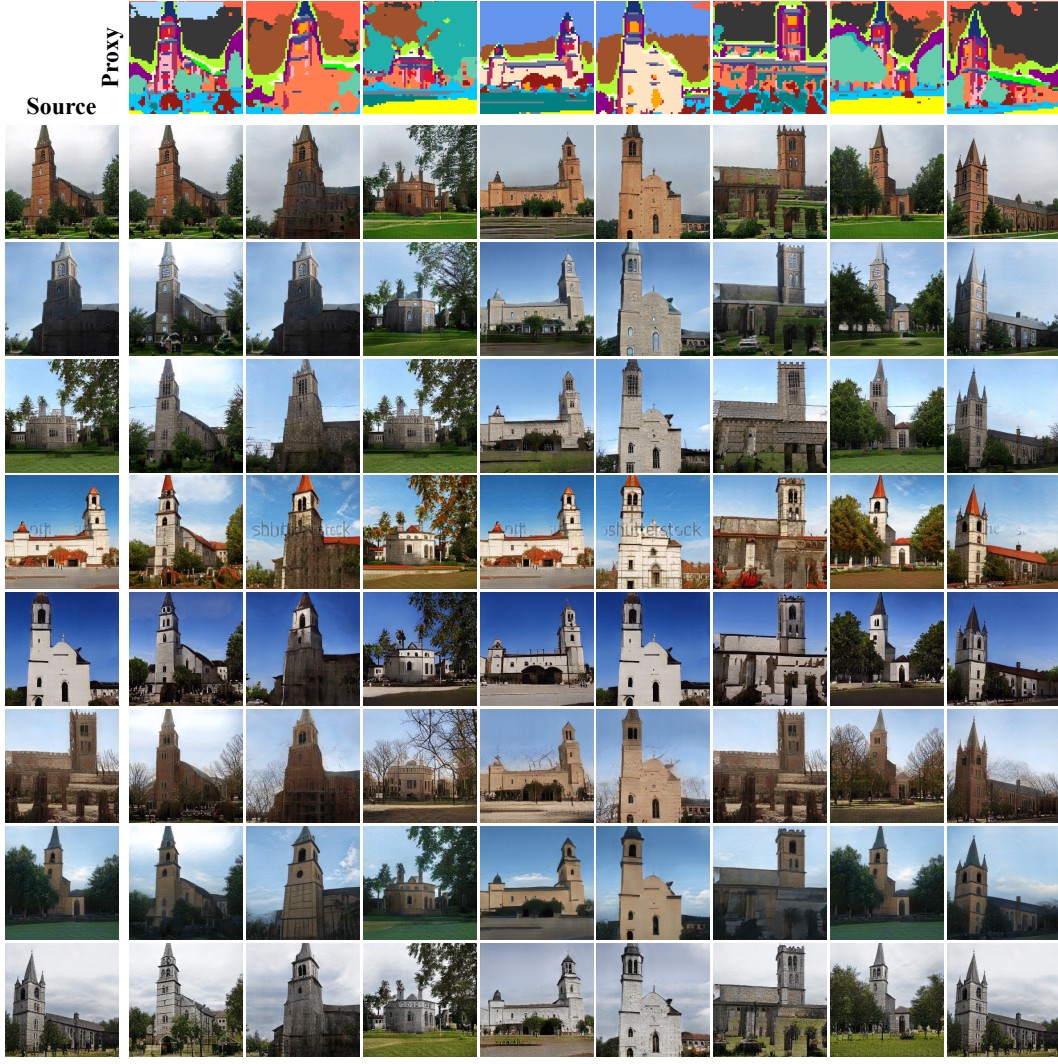

Figure S11: **Ours using the proxy mask obtained from random noise on LSUN Church dataset**
By generating images using the proxy mask without considering the perception gap, it is possible to create images of a higher quality.

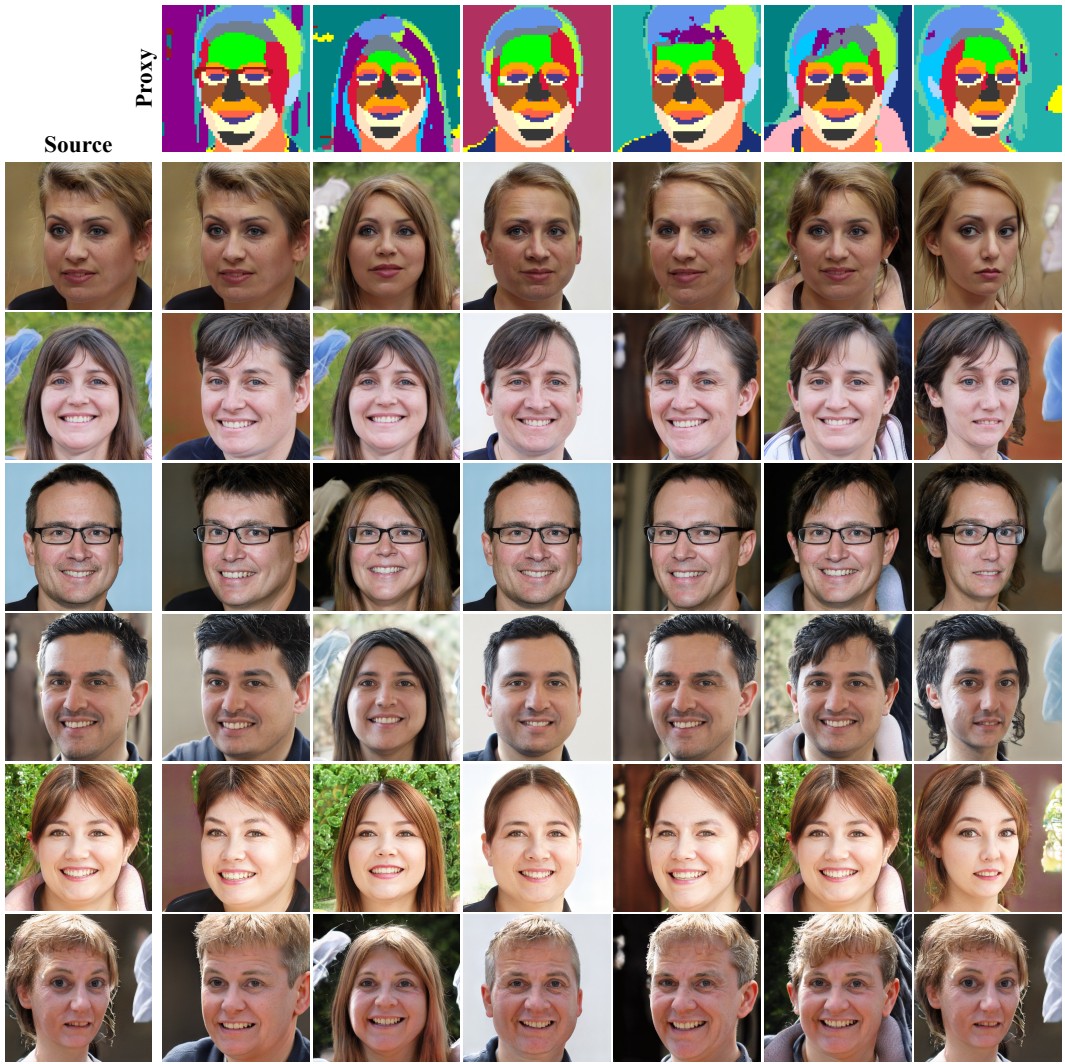

Figure S12: **Ours using the proxy mask obtained from random noise on FFHQ dataset** By generating images using the proxy mask without considering the perception gap, it is possible to create images of a higher quality.

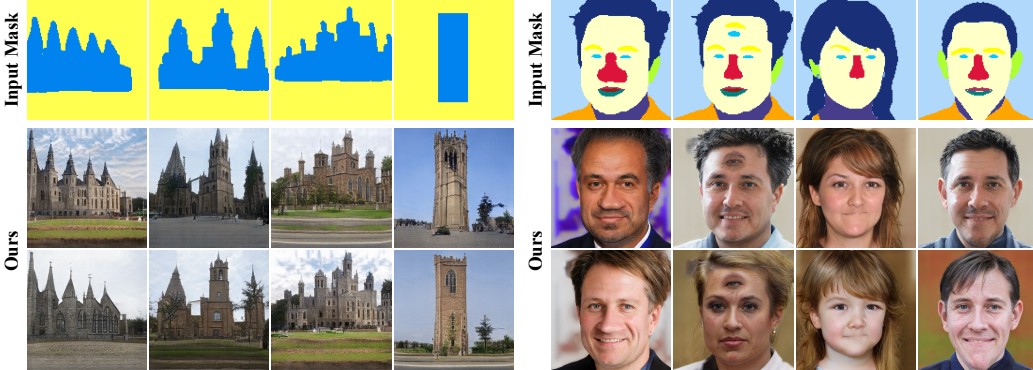

(a) **Ours reflecting free-form mask on LSUN Church**     (b) **Ours reflecting free-form mask on FFHQ**

Figure S13: **Free-form image manipulation** When given free-form mask images, our model flexibly synthesizes images. Even masks with unusual shape, such as a rectangular mask of LSUN Church or a three-eyed mask for FFHQ, can generate proper images.

# G   More free-form image manipulation

In Figure S13. We show the results using free-form image manipulation on LSUN Church and FFHQ.