# OpenReview forum: "Semantic Image Synthesis with Unconditional Generator"
_NeurIPS.cc/2023/Conference — NeurIPS 2023 poster_

### Official Review · Reviewer_4iJ7 · 2023-07-02

**Soundness:** 3 good
**Presentation:** 2 fair
**Contribution:** 3 good
**Rating:** 5
**Confidence:** 3

**Summary:**

This work proposes a novel method for semantic image synthesis from an unconditional generator. Specifically, a proxy mask is used to provide semantic guidance to reararange the middle layer's feature. A segmentation module is learned to map user input to the proxy mask in a few-shot manner. Experiments over different datasets show proposed method's effectiveness.

**Strengths:**

1. Semantic image synthesis typically requires learning from a large amount of paired data. However, this work relaxes this requirement to just a single pair. This approach is innovative and has the potential to unlock new applications that enable the control of image synthesis through customized signals.

2. The authors have implemented thorough experiments over different dataset and different kinds of control signals (seg, sketch, scribble, etc.).

**Weaknesses:**

1. The formulation is messy. For example, $m_{proxy}$ and $K$ (in Equation 2) are undefined when they are first mentioned in the paper.

2. There are many loss terms proposed. However, the authors do not provide ablation study results over them.



**Questions:**

The image quality is not as good as previous methods for full-shot setting. Why does this happen? I hope the authors could provide some analysis and hypotheses, and it would be even better if they could conduct experiments to support their claims.

**Limitations:**

The authors have discussed the limitations in the paper.

---

> ### Author Rebuttal · Authors · 2023-08-10
>
> Thank you for recognizing the advantages of our paper: While conventional methods require a large amount of paired data for training, ours is innovative in that it requires only a single pair to enable training. Furthermore, our approach has the potential to be applied to facilitate the control image synthesis.
>
> #### Expressions
> >The formulation is messy. For example, m_proxy and K (in Equation 2) are undefined when they are first mentioned in the paper.
>
>
> Thanks for pointing out the formulation, we will fix them in our revised version.
> m_proxy denotes proxy mask, $K$ denotes K-means clustering algorithm, $G_1$ is the earlier layers of the StyleGAN up to the layer corresponding to the feature maps used to generate the proxy mask. $G_2$ is the rest of the layers of the model.
>
> >There are many loss terms proposed. However, the authors do not provide ablation study results over them.
>
> The ablation study of using loss terms to train a Rearranging Network is shown in Appendix B.1 of our paper. Appendix Fig. S3 shows that reconstruction loss and mask loss play a crucial role in the training. There is an additional ablation study for Semantic Mapper loss terms below. Training with the adversarial loss while training Semantic Mapper helps to get better quality with FID 18.53, although we can get proper images without using adversarial loss. The quantitative results are in the table below.
> |        | Semantic Mapper with adv loss | Semantic Mapper without adv loss |
> |--------|----------------------------------|-------------------------------|
> | FID    | 18.53                            | 21.30                         |
> | mIoU   | 53.10                            | 55.12                         |
>
>
> #### Quality of the results
>
> >The image quality is not as good as previous methods for full-shot setting. Why does this happen? I hope the authors could provide some analysisma and hypotheses, and it would be even better if they could conduct experiments to support their claims.
>
>
> The general response 1. shows that our method performs well in one-shot training. The main goal and advantage of our method is to use pretrained generator without many annotation pairs.  The performance of one-shot is highly dependent on the annotation of a single image, so pixel-level mislabeling affects the image quality. Figure R1 in the PDF shows the corresponding failure cases. In addition, as mentioned in the general response 1, the performance increases when pretrained segmentation model is added instead of one-shot segmentation model. (FID 15.37)

---

> > ### Comment · Reviewer_4iJ7 · 2023-08-18
> >
> > Thanks for clarification. My concerns are addressed. I would like to remain my positive rating.

---

> > > ### Author Response · Authors · 2023-08-19
> > >
> > > Thank you for your recognition and for taking the time to express your feedback. We will be happy to answer any further questions or additional suggestions that may arise.

---

### Official Review · Reviewer_iJE3 · 2023-07-03

**Soundness:** 3 good
**Presentation:** 3 good
**Contribution:** 2 fair
**Rating:** 5
**Confidence:** 5

**Summary:**

In this paper, authors propose to use well pre-trained StyleGANv2 for conditional image synthesis, e.g., semantic image synthesis. To achieve conditional control, a rearranging mapping network and a semantic mapper module are proposed to connect the semantic input and middle spatial feature control within the StyleGANv2 framework. Experiments show that the proposed method can generate images under various input conditions.

**Strengths:**

1. The paper is well organized  and easy to understanding.
2. The proposed method is flexible to various conditional inputs, such as semantic mask and sketch.

**Weaknesses:**

1. As mentioned in the limitation section of the paper, the quality of generated image is not very good. For example, images (FFHQ) in Figure 5 show that the shape of hair is not well aligned with the input mask. In Figure 7, the background of cat shows artifacts and the structure of hair is not match the sketch (third column). It seems that the proposed method still can not control the spatial conditions well, although it is better than previous method, LinearGAN.
2. The ablation study can be further improved.
    a. Why self reconstruction loss and mask loss are used different format (i.e., MSE loss and CE loss)?
    b. What is the main factor for the performance of the method (self-supervised learning stage, or semantic guide stage)?  If we use full-shot segmentation network, can we further benefit the proposed method?
    c. It seems that given a semantic mask, it is first transferred to a proxy mask, and then mapped to features ($F_{arr}$) along with the $F_{ori}$ (generated with random noise). An intuitive feeling is that when the randomly generated $F_{ori}$ is similar to the condition (proxy mask), the synthesis result will be better. So is the synthesis result sensitive to random noise?
3. Some expressions are not very clear. Authors claim that existing supervised methods require pixel-level annotations and are need to be re-trained if the number of classes changes. But the proposed method still need to re-train the semantic mapper if we want to add new semantic classes. So, I'm not sure what the author wants to emphasize.
4. With the development of large-scale vision fundament models, like SAM, we can now easily get the pixel-level annotations of images. And the results in the paper show that the performance of the proposed method is not good as full-supervised methods (OASIS). Thus, I suggest to provide more detailed analysis of the advance and potential applications of the proposed method.

**Questions:**

  My questions are listed in the weaknesses.

**Limitations:**

Authors have discussed the limitations.

---

> ### Author Rebuttal · Authors · 2023-08-10
>
> Thank you for recognizing our advantages: Our method is flexible to diverse conditions.
>
> #### Quality of the results
>
> >As mentioned in the limitation section of the paper, the quality of generated image is not very good. For example, images (FFHQ) in Figure 5 show that the shape of hair is not well aligned with the input mask. In Figure 7, the background of cat shows artifacts and the structure of hair is not match the sketch (third column). It seems that the proposed method still can not control the spatial conditions well, although it is better than previous method, LinearGAN.
>
> We use masks at $64^2$ resolution and the images are $1024^2$ resolution. The gap between mask and image inevitably leads to limited precision in control such as fine hair. These are mentioned in the limitation. Similarly, the sketch in Figure 7 also becomes a proxy mask that is low resolution to generate the image. Therefore, it does not reflect a perfect alignment of hair details. Nevertheless, ours achieve much better alignment than LinearGAN and pSp. Artifacts in the background of the image occur in only a handful samples. As shown in Figure R4 in the PDF, most of the images generated with styles and scribbles as conditions still show a strong ability to reflect the given conditions and maintain high quality without artifacts.
>
> #### Ablation
> >a. Why self reconstruction loss and mask loss are used different format (i.e., MSE loss and CE loss)?
> >b. What is the main factor for the performance of the method (self-supervised learning stage, or semantic guide stage)? If we use full-shot segmentation network, can we further benefit the proposed method?
> >c. It seems that given a semantic mask, it is first transferred to a proxy mask, and then mapped to features (F_arr) along with the F_ori (generated with random noise). An intuitive feeling is that when the randomly generated is similar to the condition (proxy mask), the synthesis result will be better. So is the synthesis result sensitive to random noise?
>
> **a. Design choice of loss functions** We used cross-entropy to align pixel classes in predicted and target masks, a common loss function for semantic segmentation and image synthesis tasks involving mask prediction. However, for feature maps, it is a regression problem rather than a pixel-by-pixel classification. Therefore, we used the L2 loss accordingly.
>
> **b-1. Main factor of performance**  The main goal and advantage of our work is to perform semantic image synthesis using the pre-trained unconditional generator, rather than performance improvement. Therefore, the Rearranger, the Semantic Mapper and the training strategies are all the main factors.
>
> **b-2. Full-shot segmentation network** Table 2 also shows that performance increases with increasing annotation. Additionally, we observed a better result in FID (15.37) when using the pre-trained segmentation network with CelebA-MaskHQ instead of the one-shot annotation. When we use the mIoU metric also increased from 53.10 to 60.92.
>
>
> **c. Independence between the input and condition**
> In Appendix D. Figures S9 and S10, the diagonal direction shows images generated with feature maps and proxy masks from the same latent. The results show that the reproduction is almost identical to the original. However, when we looked at other images, we found that using feature maps and proxy masks generated from different latents did not affect the image quality. Note that this is different from giving a custom mask as a condition. For a quantitative comparison, we measured the FID and found that the images generated from random latent pairs had an FID of 7.48, while the images generated from the same latent had an FID of 9.90.
>
>
>
>
> #### Flexibility to changing conditions
>
> >Some expressions are not very clear. Authors claim that existing supervised methods require pixel-level annotations and are need to be re-trained if the number of classes changes. But the proposed method still need to re-train the semantic mapper if we want to add new semantic classes. So, I'm not sure what the author wants to emphasize.
>
>
> Our method also needs to be retrained when the number of classes changes. However, our method consists of two modules, the Rearranging Network and the Semantic Mapper. The training time for this is only **12 hours** for CelebA-MaskHQ ($256^2$), since only the latter needs to be retrained if the input conditions or the number of classes change. On the other hand, given that the SIS competitor takes 76 hours, it is more advantageous to respond to changes in input conditions, such as changes in the number of classes. (see results in Figure R3 in the PDF)
>
>
>
> #### Efficiencies of the proposed method
>
> >With the development of large-scale vision fundament models, like SAM, we can now easily get the pixel-level annotations of images. And the results in the paper show that the performance of the proposed method is not good as full-supervised methods (OASIS). Thus, I suggest to provide more detailed analysis of the advance and potential applications of the proposed method.
>
>
> As mentioned in general response 2, There are still advantages to our method when the desired semantics are outside the semantic boundaries of SAM, or when a custom mask is desired. Also, the advantage of our method lies not only in its data efficiency but also in the time efficiency for training compared to training an SIS model from scratch. Furthermore, unlike supervised SIS methods, our method is also versatile enough to generate images that reflect various types of input conditions, such as sketches.

---

> > ### Comment · Reviewer_iJE3 · 2023-08-17
> >
> > Thanks for authors' response. I'd like to improve my score.

---

> > > ### Author Response · Authors · 2023-08-18
> > >
> > > Thank you for your recognition and swift feedback. We will be happy to answer any further questions that may arise.

---

### Official Review · Reviewer_pArX · 2023-07-04

**Soundness:** 3 good
**Presentation:** 3 good
**Contribution:** 3 good
**Rating:** 4
**Confidence:** 4

**Summary:**

This paper proposed a self-supervised semantic image generation method. It adopts a pretrained generator, and rearranges the intermediate features to make the generated images align with the given input. Extensive experiments show the method achieve comparable performance with previous GAN based methods.

**Strengths:**

The training is self-supervised and does not require paired data collection.

The writing is clear.

Results on diverse datasets, including faces and buildings.

**Weaknesses:**

- The key motivation of this work is that it is difficult to collect large-scale paired data for semantic generation models. But this is not challenging for the cases shown by authors, i.e., mask-to-image, sketch-to-image. For example, we can use well-trained semantic segmentation models, e.g., SAM, to perform semantic segmentation, and use the sketch extraction method, e.g., HED, Canny, to extract sketches. In this way, we can easily obtain large-scale paired training data. Also, there are also some methods, e.g., PITI that use pretraining to alleviate the data issue.
- The visual results do not show clear improvement than previous methods in Fig. 4, 5. For sketch to image generation in Fig. 7, the generated images do not align with the given inputs.
- For k-means, how do you decide the cluster numbers? How would the noisy clustered mask affect the results? As the proxy mask is noisy, how would it achieve fine control of generated results?


**Questions:**

Are G1 and G2 in Fig. 2 the same model?

**Limitations:**

see weakness.

---

> ### Author Rebuttal · Authors · 2023-08-10
>
>
> Thank you for your comment and for acknowledging our strengths. Our method is self-supervised and doesn't need paired mask-image collection.
>
> #### Comparison with full dataset generation using semantic segmentation models(SAM)
>
> >The key motivation of this work is that it is difficult to collect large-scale paired data for semantic generation models. But this is not challenging for the cases shown by authors, i.e., mask-to-image, sketch-to-image. For example, we can use well-trained semantic segmentation models, e.g., SAM, to perform semantic segmentation, and use the sketch extraction method, e.g., HED, Canny, to extract sketches. In this way, we can easily obtain large-scale paired training data. Also, there are also some methods, e.g., PITI that use pretraining to alleviate the data issue.
>
>
> As mentioned in general response 2, There are still advantages to our method when the desired semantics are outside the semantic boundaries of SAM, or when a custom mask is desired.  Also, the use of SAM can address data-related challenges. However, using SAM requires training from scratch. In contrast, our approach uses a pre-trained model. This demonstrates not only annotation efficiency but also time efficiency compared to training a semantic image synthesis model from scratch. Furthermore, existing SIS models do not present results for the diverse range of conditions presented in our approach.
>
>
> #### Quality of the results
>
> >The visual results do not show clear improvement than previous methods in Fig. 4, 5. For sketch to image generation in Fig. 7, the generated images do not align with the given inputs
>
> In Fig. 4, OASIS (one-shot) produces faces with components from multiple people e.g., ears and clothes. However, our approach does not show such characteristics. In Fig. 5, our approach aligns better than LinearGAN. For example, compared to LinearGAN, our work aligns the detailed shape of the hair better. In addition, our work also presents a more natural appearance of the face. This is because ours can give users desired styles with the control of latents. However, since LinearGAN uses random initialization of latents and latent optimization process with cross entropy loss, it is difficult to give user preferences. Furthermore, our approach shows lower FID scores compared to previous work. In Fig. 7, when comparing the shape of the hairstyle across all result images, our approach aligns with the sketch better than pSp.
>
> #### How to determine the number of clusters
>
> >For k-means, how do you decide the cluster numbers?
>
> Increasing the number of clusters improves image quality and fine control. However, employing excessively many numbers of clusters reduces consistency between the proxy mask and human understanding, making semantic mapper hard to train. Hence, we opt for 25 clusters as an optimal choice. Appendix B.2 provides the above experiments.
>
>
> #### The proxy mask is noisy
> >  How would the noisy clustered mask affect the results? As the proxy mask is noisy, how would it achieve fine control of generated results?
>
> We interpreted the term "noisy clustered mask" to mean that the proxy mask contains some noise. Even if the proxy mask is somewhat noisy, adversarial training favors the output image to be in the domain. However, the model also learns to accurately reflect the proxy mask in the image through $\mathcal{L}_{\text{mask}}$. Therefore, if the noise is too strong, it may corrupt the image. If our interpretation of the term "noisy clustered mask" is incorrect, we would appreciate additional context to clarify its meaning.
>
> #### G1 and G2
> >Are G1 and G2 in Fig. 2 the same model?
>
> FullGenerator($z$) = $G_2(G_1(z))$, where FullGenerator denotes the entire generator of StyleGAN2. In other words, $G_1$ is the composite of layers earlier than the rearranger. $G_2$ is the rest.

---

> ### Comment · Area_Chair_4ggQ · 2023-08-18
>
> Reviewer pArX,
>
> Please read the rebuttal provided by authors and raise a discussion if your concerns are not well addressed.
>
> Best,
> AC

---

> ### Comment · Reviewer_pArX · 2023-08-18
>
> Thanks to the authors for their responses. After reading the rebuttal and comments from other reviewers,I am still concerned about the effect of noisy masks. The inaccurately clustered masks are quite common in the paper's setting due to lack of supervision. This is not a trivial problem, but it is a key challenge in the proposed method. But the current submission does not propose specific designs to address this issue, and how the noisy masks would affect the generation result is unclear. The resulting noisy generation may not be fully reflected in quantitative metrics like PSNR, as the noisy regions can be relatively small. But the visual results, in this case, can show obvious artifacts, which would hinder the application of the proposed method. Therefore, I would keep my original ratings.

---

> > ### Author Response · Authors · 2023-08-19
> >
> > First of all, we appreciate your constructive feedback.
> >
> > > The inaccurately clustered masks are quite common in the paper's setting due to lack of supervision.
> >
> > While the mask is slightly noisy due to the unsupervised settings, previous work and we have focused on the clustering results are accurately segmented for semantics [F].
> >
> >
> > [F]  Collins, et al. Editing in Style: Uncovering the Local Semantics of GANs. CVPR 2020.
> >
> >
> > > It is a key challenge in the proposed method.
> > But the visual results, in this case, can show obvious artifacts, which would hinder the application of the proposed method.
> >
> >
> > Figures S8, S9, and S10 show the results of the images generated with the noisy mask. The slight noisiness of the proxy mask has little impact on generating images that reflect input conditions such as segment masks and sketches, so we don't see it as a key challenge. If there is a concern regarding a specific figure, it would be helpful if you could point it out.
> >
> >
> > > But the current submission does not propose specific designs to address this issue,
> >
> > $L_{\text{self}}$ and $L_{\text{mask}}$ allow the proxy mask to be accurately reflected in the rearranged feature maps. As a complement to these, $L_{\text{adv}}$ not only improves the quality of the image, but also makes it robust to noisy conditions.
> >
> >
> > > how the noisy masks would affect the generation result is unclear
> >
> > Referring back to Figures S8, S9, and S10, the proxy mask entering the condition is quite noisy, but it produces good results.

---

### Official Review · Reviewer_swNj · 2023-07-04

**Soundness:** 4 excellent
**Presentation:** 4 excellent
**Contribution:** 3 good
**Rating:** 6
**Confidence:** 5

**Summary:**

The paper introduces a method to adapt unconditional GAN generators to pixel-level-conditioned tasks, such as semantic image synthesis. The method is based on the observation that the intermediate generator's features contain rich knowledge about the semantics of generated objects. This knowledge is extracted to build a proxy mask via clustering, which can be manipulated by the introduced Reanranging Network. The manipulation to a user-specified conditioning mask is performed by a separate component called Semantic Mapper. Overall, the method allows to achieve high-quality, highly conrollable SIS without large paired image-mask datasets. The experiments also demonstrate a range of applications, including image manipulation, style-based SIS, and sketch-to-image synthesis.

**Strengths:**



[Contribution] - The main strength, in my opinion, is itself the successful SIS from unconditional GANs. The literature on SIS GANs has long been somewhat detached from unconditional GANs, with its own tricks and training methods (e.g., pix2ix architectures, perceptual losses) and shortcomings (e.g., limited diversity). Apart from the data efficiency (claimed by authors), applying unconditional GANs to SIS brings the potential to improve SIS with all the recent advancements from other GAN literature.

[Efficiency] - The proposed method uses a pre-trained GAN generator and one manual annotation (proxy mask -> real mask). This is a very cheap method to achieve SIS compared to conventional SIS GAN models. This can enable SIS on datasets that do not have good segmentation datasets, which can reduce costs. As the authors claimed, if the segmentation style changes, this model does not need to be re-trained, just the Semantic Mapper.

[Performace] - Unconditional synthesis is naturally more difficult than SIS with paired datasets, so the quality of the model does not reach all prior SIS GANs trained on full datasets. But it is impressively not far, both in FID and mIoU.

[Applications] - Some applications enabled by the model seem to extend the prior capabilities of both unconditional GANs and SIS GANs.

**Weaknesses:**

[Limitation] - Enforcing each generated image to have K classes via K-means clustering seems rather restrictive. This works for FFHQ and Churches, but would not work for more classic SIS datasets like Ade20K, Cityscapes, or Coco-Stuff, where each image has only a subset of semantic classes. Solving this requires, probably, nontrivial extensions.

[Evaluation] - One point of using unconditional GANs instead of SIS could be their better conditioning and diversity. I think it would be interesting to see more analysis on unconditional GANs vs SIS GANs, to see the trade-offs. For example, is LPIPS diversity of synthesis higher with StyleGANv2 compared to, say, OASIS? Other metrics like PPL, Precision&Recall would be also interesting.

[Semantic Mapper] - It is a bit unclear how Semantic Mapper is trained. Is it correct that for a single given proxy mask, the user has to draw a real mask himself? In this case, it is also somehow a limitation, as it introduces overhead.  More research is needed to find proxy-real mask alignment more automatically.


**Questions:**

Please answer the concerns in Weaknesses.



**Limitations:**

No, more discussion is needed. I suggest adding the limitation points from Weaknesses if they remain relevant after the rebuttal. There is no discussion on societal impact at all.

---

> ### Author Rebuttal · Authors · 2023-08-10
>
> We appreciate your thorough recognition of the strengths in our paper: Our method offers the advantage of applying unconditional GANs to SIS. Furthermore, when the segmentation style changes, only retraining the Semantic Mapper is required, showcasing another strength. The model enables new applications that appear to expand the capabilities of both unconditional GANs and SIS.
>
> #### Explanation of feature maps clustering
> >Enforcing each generated image to have K classes via K-means clustering seems rather restrictive. This works for FFHQ and Churches, but would not work for more classic SIS datasets like Ade20K, Cityscapes, or Coco-Stuff, where each image has only a subset of semantic classes. Solving this requires, probably, nontrivial extensions.
>
> We cluster the set of features from *512 samples* rather than each sample to K clusters. Formally,  $\mathbf{f}_i = G_1(z_i)$ where $z_i$ is a random latent code and $\mathbf{f}_i \in \mathcal{R}^{CHW}$  are feature maps. We run K-means over a set of features $\mathbf{f}_i(x,y)$ to prepare K cluster centroids where $x$ and $y$ are coordinates of feature maps $\mathbf{f}_i$. Hence, our method is applicable to such datasets if their unconditional generators are publicly available. We emphasize that our method enriches the boundary of SIS to datasets without semantic masks.
>
>
> #### Diversity of proposed method
>
> >One point of using unconditional GANs instead of SIS could be their better conditioning and diversity. I think it would be interesting to see more analysis on unconditional GANs vs SIS GANs, to see the trade-offs. For example, is
> diversity of synthesis higher with StyleGANv2 compared to, say, OASIS? Other metrics like PPL, Precision&Recall would be also interesting.
>
> Thank you for the insightful comment. Yes, the diversity of our method measured by LPIPS is higher than INADE and OASIS. Consulting precision and recall together does not pick a definite winner. The table below summarizes the measures. For measuring LPIPS, we follow INADE [E].
> |           | Ours | INADE | OASIS |
> |-----------|------|-------|-------|
> | LPIPS     | 0.45 | 0.36  | 0.30  |
> | Precision | 0.76 | 0.86  | 0.69  |
> | Recall    | 0.22 | 0.19  | 0.36  |
>
>
> [E] Tan, Zhentao, et al. Diverse semantic image synthesis via probability distribution modeling. CVPR 2021.
>
> #### Training method of the Semantic Mapper and Advantages of self-made Mask
>
> >It is a bit unclear how Semantic Mapper is trained. Is it correct that for a single given proxy mask, the user has to draw a real mask himself? In this case, it is also somehow a limitation, as it introduces overhead. More research is needed to find proxy-real mask alignment more automatically
>
>
>
> The semantic mapper is a function that receives a semantic mask and produces a proxy mask. It requires pairs of semantic masks and proxy masks for training. For a random fake image and its proxy mask produced by $G_2(G_1(z))$ and $G_1(z)$, respectively, a user draws a semantic mask for the fake image (yes it is correct). By letting the user draw the mask, our method allows different types of inputs ranging from the custom number of semantic classes to binary contour scribbles according to the intention of user.

---

> > ### Comment · Reviewer_swNj · 2023-08-14
> > **Answer from Reviewer swNj**
> >
> > I thank the authors for their answers.
> >
> > My questions were addressed adequately, and, in addition, Appendix B.2 helped to reduce my confusion about the clustering.
> >
> > I remain with my positive evaluation and have no further questions for the authors at this point.

---

> > > ### Author Response · Authors · 2023-08-15
> > >
> > > Thank you for your recognition and swift feedback. We will be happy to answer any further questions that may arise.

---

### Official Review · Reviewer_Lq4s · 2023-07-06

**Soundness:** 3 good
**Presentation:** 2 fair
**Contribution:** 3 good
**Rating:** 5
**Confidence:** 5

**Summary:**

This paper proposes a new method for semantic image synthesis that can generate images from given semantic segmentation maps, particularly in few-shot settings with different forms of inputs. The method trains a model to transform the input semantic segmentation maps into feature maps and cluster the output to outline the discriminativeness and boundaries in the resulting features. It then learns a conditional GAN to generate realistic images conditioned on random noise and the processed feature maps transformed from the corresponding semantic layouts. These designs enable few-shot capability and handle various forms of inputs (such as sketches and scribbles). The proposed method is extensively tested in experiments, which validate its effectiveness.

**Strengths:**

1. The paper proposes an interesting method design that enables few-shot image synthesis by predicting a proxy mask from intermediate features predicted from the input. This approach also enables the method to handle inputs in different forms beyond just segmentation maps.
2. The experiments conducted on CelebAMask-HQ, LSUN Church, and LSUN Bedroom datasets validate the effectiveness of the proposed method. The one-shot performance on FID, mIoU, and accuracy metrics is impressive.

**Weaknesses:**

1. The paper should discuss why certain method designs were not ablated, such as the use of k-means to generate the proxy mask from predicted features. The impact of this design choice on the effectiveness of the proposed method should also be measured and discussed.
2. The paper should include a discussion of the capacity and efficiency of the proposed model. Specifically, it should analyze how scaling the model impacts the quantitative results.
3. To provide a more comprehensive understanding of the proposed method, the paper should include an analysis of failure cases and discuss their limitations.

**Questions:**

1. It would be interesting to investigate whether the semantic mapper could be initialized by pretrained semantic segmentation models or how its design could be improved. This could potentially improve the final performance and is worth exploring further.
2. Since diffusion-based methods can now be used for semantic segmentation, it raises the question of whether this framework could be extended to diffusion-based image synthesis. This is an intriguing possibility to explore, although there may be potential difficulties to overcome, such as the complexity of the model and the computational resources required.

**Limitations:**

To explore the method's limitations, it would be beneficial to discuss some failure cases. This could help identify situations where the proposed method may not perform well and provide insight into potential areas for improvement.

---

> ### Author Rebuttal · Authors · 2023-08-10
>
> Thank you for recognizing our positive aspects: Ours has an interesting method design and showcases impressive one-shot performance across FID, mIoU, and accuracy metrics.
>
> ### Ablation
> >The paper should discuss why certain method designs were not ablated, such as the use of k-means to generate the proxy mask from predicted features. The impact of this design choice on the effectiveness of the proposed method should also be measured and discussed
>
> The key components of our method are rearranger, semantic mapper, and the training scheme. We follow existing works for peripheral components as follows. We use k-means because it is a common choice for clustering [A]. We also chose cross attention, which is a technique commonly used to incorporate conditions into inputs, such as transforming references into the form of the target [B,C].
>
> [A] Collins et al. Editing in Style: Uncovering the Local Semantics of GANs. CVPR 2020.
> [B] Liu et al. DynaST: Dynamic Sparse Transformer for Exemplar-Guided Image Generation. ECCV 2022.
> [C] Rombach et al. High-Resolution Image Synthesis with Latent Diffusion Models. CVPR 2022.
>
>
> #### Capacity, efficiency, and scaling
> >The paper should include a discussion of the capacity and efficiency of the proposed model. Specifically, it should analyze how scaling the model impacts the quantitative results
>
> We performed the scaling experiment of the proposed module indirectly by reducing the original CNN layer by half. For the semantic mapper, we have not experimented with further scaling, as it is identical to the generator of OASIS except that it does not use 3D noise, and we will experiment further if there are questions about this part. The details of the structure are described in Appendix A.1. The results of the experiments are shown in the table below. We report the number of parameters (capacity), training time (efficiency), and FID & mIoU over different scales (scaling). The results show that our method is time efficient to train and achieves high image quality. The efficiency of our module comes from the fact that it only learns to rearrange the representation, unlike existing SIS methods.
>
> |                                                          	| number of params 	| training time 	| FID   	| mIoU 	|
> |----------------------------------------------------------|------------------|---------------|-------|------|
> | Rearranging network (Number of CNN layers halved) 	| 30M           	| 8 hours        	|  8.71 	| n/a  	|
> | Rearranging network                                  	| 54M           	| 18 hours       	|  7.48 	| n/a  	|
> | Semantic Mapper                                      	| 69M           	| 12 hours       	| 18.53 	| 53.10 |
> | OASIS                                                 | 93M           	| 77 hours       	| 20.23 	| 74.01 |
>
>
>
>
>
>
> #### Failure cases
>
> >To provide a more comprehensive understanding of the proposed method, the paper should include an analysis of failure cases and discuss their limitations.
>
>
> We show the failure case in Figure R1. The result shows that the mask does not reflect well in some areas, and there are two reasons for this. First, as mentioned in Limitation, the proxy mask has a lower resolution of $64^2$ than the actual resolution of $256^2$. Therefore, some semantics are lost when mapping to the proxy mask. The next case of failure is a mislabeled annotation. Since the semantic mapper is completely dependent on a single mask, we could observe cases where, for example, if a strand of hair is not correctly annotated, the effect is reflected in the result, and the hair becomes the background. Furthermore, these cases improve with increasing annotation, as shown quantitatively in Table 2.
>
>
>
>
> #### Weight initialization using pretrained segmentation models
> >It would be interesting to investigate whether the semantic mapper could be initialized by pretrained semantic segmentation models or how its design could be improved. This could potentially improve the final performance and is worth exploring further
>
>
> It is indeed an interesting idea. However, semantic mapper receives semantic masks, while semantic segmentation models receive RGB images. Initializing weights from a pretrained model usually replaces last layers. On the other hand, adapting a pretrained segmentation model for semantic mapper requires replacing the input layer.  Thank you for initiating great discussion.
>
>
> #### Extension to the diffusion models
> >Since diffusion-based methods can now be used for semantic segmentation, it raises the question of whether this framework could be extended to diffusion-based image synthesis. This is an intriguing possibility to explore, although there may be potential difficulties to overcome, such as the complexity of the model and the computational resources required.
>
>
> We agree that our method is applicable for diffusion models. We first confirmed that the proxy mask can be obtained by clustering the latents of the U-net of stable diffusion, and there is also related work on clustering with latents [D]. Manually modifying the latents in a single timestep successfully edits the resulting image. We have edited facial photos to have three eyes, and the result is attached as figure R2 in the PDF. We have tried to add a Rearranging Network and a Semantic Mapper to the model. However, it was not straightforward to make it work. This makes it a plausible next research item.
>
> [D] Xu et al. ODISE: Open-Vocabulary Panoptic Segmentation with Text-to-Image Diffusion Models. CVPR 2023.

---

> > ### Author Response · Authors · 2023-08-19
> >
> > Dear reviewer Lq4s
> >
> > We are deeply appreciate your insightful comments and suggestions on our paper. In response to your comments, we have conducted further experiments and submitted our rebuttal accordingly. It would be grateful if you could let us know your thoughts on our responses. We would be happy to answer additional questions if any.
> >
> > Sincerely,  Authors

---

### Author Rebuttal · Authors · 2023-08-10

# Global Response
We thank all the reviewers for their insightful reviews. We first summarize the strengths of our paper recognized by the reviewers.


1. Our method can apply unconditional GANs to SIS, which has the potential to leverage recent advancements from other GAN literature.
2. The results demonstrate the flexibility of our approach to handle various conditional inputs and its applicability to diverse datasets.
3. Training process is self-supervised, eliminating the need for paired data collection.
4. Ours have been conducted thorough experiments over different datasets and control signals, further validating the effectiveness of our approach.
5. One-shot performance on FID, mIoU, and accuracy metrics is impressive, showcasing its effectiveness in various evaluation aspects.


Next, we aim to address common concerns raised by reviewers.

**1. Comparison with supervised method in image quality**(pArX, iJE3, 4iJ7)
First, our method focuses on one-shot learning, so it is not fair to compare it with a fully annotated supervised method. For our targeted one-shot learning, our method outperforms competitors, including supervised methods. However, due to extreme annotation limitations, the efficacy of the mapper might be constrained.
Therefore, we conducted additional experiments to confirm whether increased supervision improves performance. For the experiments, we utilized a pretrained segmentation network for CelebA-MaskHQ instead of one-shot segmentation network. All other models are the same. In this setting, as in the case of using sketches as an input condition, the output of the generated image passed through Unet and the proxy mask made from the feature maps corresponding to the image were used as a training pair to train the semantic mapper. Consequently, the FID improved to 15.37, indicating enhanced the image quality.

**2. Pretrained vision models addressing annotation problem** (pArX, iJE3)
There have been opinions on how pretrained segmentation models like SAM can conveniently address annotation issues.
Although SAM covers many domains, it cannot cover all the semantics in different target domains. On the other hand, our method allows the use of a customized mask that contains the desired semantic information from an example image.   In addition, there is a gap with human drawings such as HED and Canny, which can be overcome by using drawings as annotations. In this way, we can use directly the user drawings as conditions when generating images.
Despite the limitation of SAM, utilizing it can facilitate the acquisition of a large paired dataset. However, conventional methods using lots of mask-image pair datasets need to require training models from scratch using the acquired dataset, leading to computation challenges. In contrast, our approach eliminates the need for from-scratch training, providing not only data efficiency but also computational efficiency. Further detailed comparisons between from-scratch and our method can be found below.

---

### Author Response · Authors · 2023-08-16

We thank the reviewers for carefully assessing our paper. We would like to ask the reviewers to engage the discussion upon our initial rebuttal.

Here is a brief reminder of our contributions.
1. Our method enables SIS with unconditional GAN.
2. We show that our method can take various input conditions not only segmentation masks but also user scribbles (different from edges) and depth maps.
3. Self-supervised training enables high-quality image generation with a data-efficient manner.
4. Compared to the existing SIS methods, adding a simple module significantly reduces the training.

We would greatly appreciate further criticisms and inquiries.

---

### Author Response · Authors · 2023-08-22

Dear all reviewers,

We deeply appreciate the time and commitment that reviewers have dedicated to reviewing our submission. The feedback received has been instrumental in enhancing our paper. We hope our responses provide clarity and aid in final assessment.

To sum up, please find our key contributions highlighted in the global response below.

Sincerely,
Authors

---

### Decision · Program_Chairs · 2023-09-21

**Decision:**

Accept (poster)

**Comment:**

Overall, the paper received positive feedbacks. The writing is clear and the proposed framework enables SIS with unconditional GAN with the help of proxy masks prediction. Authors provided additional ablations and evaluations to resolve most of the concerns raised by reviewers. Please revise the final manuscript accordingly upon acceptance.